# Value signals guide abstraction during learning

**Aurelio Cortese[1,2]\*, Asuka Yamamoto[1,3], Maryam Hashemzadeh[4], Pradyumna Sepulveda[2], Mitsuo Kawato[1,5], Benedetto De Martino[2]\***

[1]Computational Neuroscience Labs, ATR Institute International, Kyoto, Japan; [2]Institute of Cognitive Neuroscience, University College London, London, United Kingdom; [3]School of Information Science, Nara Institute of Science and Technology, Nara, Japan; [4]Department of Computing Science, University of Alberta, Edmonton, Canada; [5]RIKEN Center for Artificial Intelligence Project, Kyoto, Japan

**Abstract** The human brain excels at constructing and using abstractions, such as rules, or concepts. Here, in two fMRI experiments, we demonstrate a mechanism of abstraction built upon the valuation of sensory features. Human volunteers learned novel association rules based on simple visual features. Reinforcement-learning algorithms revealed that, with learning, high-value abstract representations increasingly guided participant behaviour, resulting in better choices and higher subjective confidence. We also found that the brain area computing value signals – the ventromedial prefrontal cortex – prioritised and selected latent task elements during abstraction, both locally and through its connection to the visual cortex. Such a coding scheme predicts a causal role for valuation. Hence, in a second experiment, we used multivoxel neural reinforcement to test for the causality of feature valuation in the sensory cortex, as a mechanism of abstraction. Tagging the neural representation of a task feature with rewards evoked abstraction-based decisions. Together, these findings provide a novel interpretation of value as a goal-dependent, key factor in forging abstract representations.

**\*For correspondence:** cortese.aurelio@gmail.com (AC); benedettodemartino@gmail.com (BDM)

**Competing interests:** The authors declare that no competing interests exist.

## Introduction

### 'All art is an abstraction to some degree.' Henry Moore

Art is one of the best examples of abstraction, the unique ability of the human mind to organise information beyond the immediate sensory reality. Abstraction is by no means restricted to high-level cognitive behaviour such as art creation. It envelops every aspect of our interaction with the environment. Imagine that you are hiking in a park, and you need to cross a stream. Albeit deceptively simple, this scenario already requires the processing of a myriad visual (and auditory, etc.) features. For an agent that operates directly on each feature in this complex sensory space, any meaningful behavioural trajectory (such as crossing the stream) would quickly involve intractable computations. This is well exemplified in reinforcement learning (RL), where in complex and/or multi-dimensional problems, classic RL algorithms rapidly collapse (*Bellman, 1957*; *Kawato and Samejima, 2007*; *Sutton and Barto, 1998*). If, on the other hand, the agent is able to first 'abstract' the current state to a lower dimensional manifold, representing only relevant features, behaviour becomes far more flexible and efficient (*Ho et al., 2019*; *Konidaris, 2019*; *Niv, 2019*). Attention (*Farashahi et al., 2017*; *Leong et al., 2017*; *Niv et al., 2015*), and more generally, the ability to act upon subspaces, concepts or abstract representations has been proposed as an effective solution to overcome computational bottlenecks arising from sensory-level operations in RL (*Cortese et al., 2019*; *Hashemzadeh et al., 2019*; *Ho et al., 2019*; *Konidaris, 2019*; *Wikenheiser and Schoenbaum, 2016*). Abstractions can be thus thought of as simplified maps carved from

higher dimensional space, in which details have been removed or transformed, in order to focus on a subset of interconnected features, that is, a higher order concept, category or schema (*Gilboa and Marlatte, 2017*; *Mack et al., 2016*).

How are abstract representations constructed in the human brain? For flexible deployment, abstraction should depend on task goals. From a psychological or neuroeconomic point of view, task goals generally determine what is valuable (*Kobayashi and Hsu, 2019*; *Liu et al., 2017*; *McNamee et al., 2013*), such that if one needs to light a fire, matches are much more valuable than a glass of water. Hence, we hypothesised that valuation processes are directly related to abstraction.

Value representations have been linked to neural activity in the ventromedial prefrontal cortex (vmPFC) in the context of economic choices (*McNamee et al., 2013*; *Padoa-Schioppa and Assad, 2006*). More recently, the role of the vmPFC has also been extended to computation of confidence (*De Martino et al., 2013*; *Gherman and Philiastides, 2018*; *Lebreton et al., 2015*; *Shapiro and Grafton, 2020*). While this line of work has been extremely fruitful, it has mostly focused on the hedonic and rewarding aspect of value instead of its broader functional role. In the field of memory, a large corpus of work has shown that the vmPFC is crucial for formation of schemas or conceptual knowledge (*Constantinescu et al., 2016*; *Gilboa and Marlatte, 2017*; *Kumaran et al., 2009*; *Mack et al., 2016*; *Tse et al., 2007*), as well as generalisations (*Bowman and Zeithamova, 2018*). The vmPFC also collates goal-relevant information from elsewhere in the brain (*Benoit et al., 2014*). Considering its connectivity pattern (*Neubert et al., 2015*), the vmPFC is well suited to serve a pivotal function in the circuit that involves the hippocampal formation (HPC) and the orbitofrontal cortex (OFC), dedicated to extracting latent task information and regularities important for navigating behavioural goals (*Niv, 2019*; *Schuck et al., 2016*; *Stachenfeld et al., 2017*; *Viganò and Piazza, 2020*; *Wilson et al., 2014*). Thus, the aim of this study is twofold: (i) to demonstrate that abstraction emerges during the course of learning, and (ii) to investigate how the brain, and specifically the vmPFC, uses valuation upon low-level sensory features to forge abstract representations.

To achieve this, we leveraged a task in which human participants repeatedly learned novel association rules, while their brain activity was recorded with fMRI. Reinforcement learning (RL) modelling allowed us to track participants' valuation processes and to dissociate their learning strategies (both at the behavioural and neural levels) based on the degree of abstraction. Participants' confidence in having performed the task well was positively correlated with their ability to abstract. In a second experiment, we studied the causal role of value in promoting abstraction through directed effect in sensory cortices. To anticipate our results, we show that the vmPFC and its connection to the visual cortex construct abstract representations through a goal-dependent valuation process that is implemented as top-down control of sensory cortices.

## Results

### Experimental design

The goal of the learning task was to present a problem that could be solved according to two strategies, based on the sampled task-space dimensionality. A simple, slower strategy akin to pattern recognition, and a more sophisticated one that required abstraction to use the underlying structure. Participants (N = 33) learned the fruit preference of pacman-like characters formed by the combination of three visual features (colour, mouth direction, and stripe orientation, *Figure 1A–B*). The preference was governed by a combination of two features, selected randomly by our computer program for each block (*Figure 1A–B*). Learning the block rules essentially required participants to uncover hidden associations between features and fruits. Although participants were instructed that one feature was irrelevant, they did not know which. A block ended when a sequence of 8–12 (randomly set by our computer program) correct choices was detected or upon reaching its upper limit (80 trials). Variable stopping criteria were used to prevent participants from learning that a fixed sequence was predictive of block termination. During each trial, participants could see the outcome after selecting a fruit. A green box appeared around the chosen fruit if the choice was correct (red otherwise). Additionally, participants were instructed that the faster they learned a block rule, the higher the reward. At the end of the session, a final monetary reward was delivered, commensurate with participant performance (see Materials and methods). Participants failing to learn the association in three blocks or more (i.e. reaching a block limit of 80 trials without having learned the

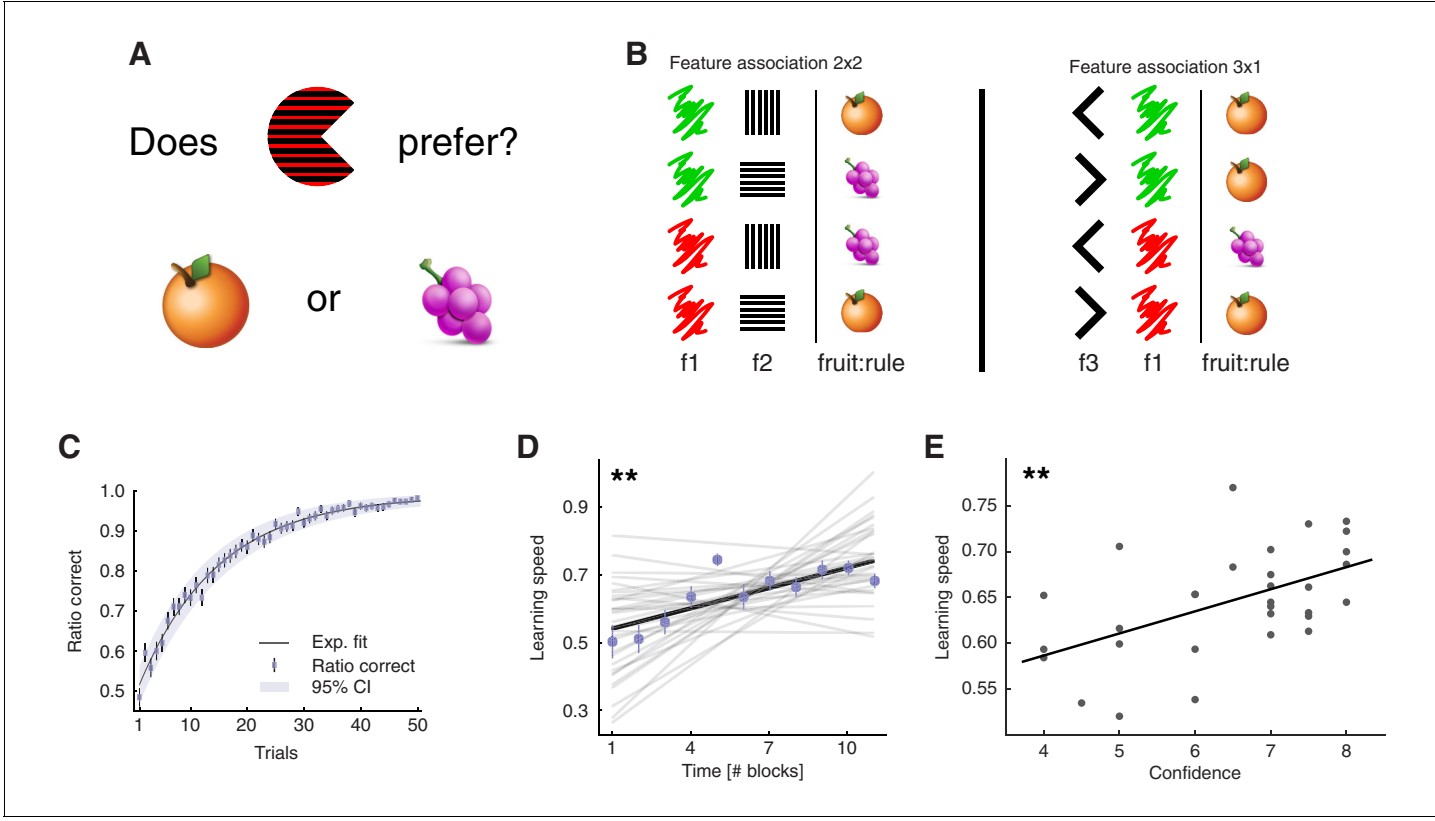

**Figure 1.** Learning task and behavioural results. (A) Task: participants learned the fruit preferences of pacman-like characters, which changed on each block. (B) Associations could form in three ways: colour – stripe orientation, colour – mouth direction, and stripe orientation – mouth direction. The left-out feature was irrelevant. Examples of the two types of fruit associations. The four combinations arising from two features with two levels were divided into symmetric (2x2) and asymmetric (3x1) cases. f1-3: features 1 to 3; fruit:rule refers to the fruit as being the association rule. Both block types were included to prevent participants from learning rules by simple deduction. If all blocks had symmetric association rules and participants knew this, they could simply learn one feature-fruit association (e.g. green-vertical), and from there deduce all other combinations. Both the relevant features and the association types varied on a block-by-block basis. (C), Trial-by-trial ratio-correct improved as a measure of within-block learning. Dots represent the mean across participants, while error bars indicate the SEM, and the shaded area represents the 95% CI (N = 33). Participant-level ratio correct was computed for each trial across all completed blocks. Source data is available in file *Figure 1—source data 1*. (D), Learning speed was positively correlated with time, among participants. Learning speed was computed as the inverse of the max-normalised number of trials taken to complete a block. Thin gray lines represent least square fits of individual participants, while the black line represents the group-average fit. The correlation was computed with group-averaged data points (N = 11). Average data points are plotted as coloured circles, the error bars are the SEM. (E), Confidence judgements were positively correlated with learning speed, among participants. Each dot represents data from one participant, and the thick line indicates the regression fit (N = 31 [2 missing data]). The experiment was conducted once (n = 33 biologically independent samples), **p<0.01.
The online version of this article includes the following source data and figure supplement(s) for figure 1:

**Source data 1.** Csv: panel C.
**Source data 2.** Csv: panel D.
**Source data 3.** Csv: panel E.
**Figure supplement 1.** Small (non-significant trends) influence of block/association type on learning speed.
**Figure supplement 2.** Behaviour analysis of excluded participants.

association), and / or failing to complete more than 10 blocks in the allocated time, were excluded (see Materials and methods). All main results reported in the paper are from the included sample of N = 33 participants.

## Behavioural accounts of learning

We verified that participants learned the task sensibly. Within blocks, performance was higher than chance as early as the second trial (*Figure 1C*, one-sample t-test against mean of 0.5, trial 2: $t_{32}$ = 4.13, $P_{(FDR)} < 10^{-3}$, trial 3: $t_{32}$ = 2.47, $P_{(FDR)}$ = 0.014, all trials t>3: $P_{(FDR)} < 10^{-3}$). Considering the

whole experimental session, learning speed (i.e. how quickly participants completed a given block) increased significantly across blocks (*Figure 1D*, N = 11 time points, Pearson's $r$ = 0.80, p = 0.003). These results not only confirmed that participants learned the task rule in each block, but also that they learned to use more efficient strategies. Notably, in this task, the only way to solve blocks faster was by using the correct subset of dimensions (the abstract representation). When at the end of a session, participants were asked about their degree of confidence in having performed the task well, their self-reports correlated with their learning speed (N = 31 [2 missing data], robust regression slope = 0.024, $t_{29}$ = 3.27, p = 0.003, *Figure 1E*), but not with the overall number of trials, or the product of the proportion of successes (learning speed: Pearson's $r$ = 0.53, p = 0.002, total trials: $r$ = −0.13, p = 0.47, test for difference in $r$: z = 2.71, p = 0.007; product of the proportion of successes: $r$ = −0.06, p = 0.75, test for difference in $r$: z = 2.43, p = 0.015). We also confirmed that the block type (defined by relevant features, e.g., colour-orientation) or association type (e.g. symmetric 2x2) did not systematically affect learning speed (*Figure 1—figure supplement 1*). Excluded participants (see Materials and methods) had overall lower performance (*Figure 1—figure supplement 2*), although some had comparable ratios correct.

## Discovery of abstract representations

Was participants' learning behaviour guided by the selection of accurate representations? To answer this question, we built upon a classic RL algorithm (Q-learning) (*Watkins and Dayan, 1992*) in which state-action value functions (beliefs) used to predict future rewards, are updated according to the task state of a given trial and the action outcome. In this study, task states were defined by the number of feature combinations that the agent may track; hence, we devised algorithms that differed in their state-space dimensionality. The first algorithm, called Feature RL, explicitly tracked all combinations of three features, $2^3$ = eight states (*Figure 2A*, top left). This algorithm is anchored at a low feature level because each combination of the three features results in a unique fingerprint – one simply learns direct pairings between visual patterns and fruits (actions). Conversely, the second algorithm, called Abstract RL, used a more compact or *abstract state representation* in which only two features are tracked. These compressed representations reduce the explored state-space by half, $2^2$ = four states (*Figure 2A*, top right). Importantly, in this task environment as many as three Abstract RL in parallel were possible, one for each combination of two features.

The above four RL algorithms were combined in a mixture-of-experts architecture (*Frank and Badre, 2012*; *Jacobs et al., 1991*; *Sugimoto et al., 2012*), *Figure 2B* and Materials and methods. The key intuition behind this approach was that at the beginning of a new block, the agent did not know which abstract representation was correct (i.e., which features were relevant). Thus, the agent needed to learn which representations were most predictive of reward, so as to exploit the best representation for action selection. Experts here denote the four learning algorithms (Feature RL, and three options of Abstract RL). While all experts competed for action selection, their learning uncertainty (RPE: reward prediction error) determined the strength in doing so (*Doya et al., 2002*; *Sugimoto et al., 2012*; *Wolpert and Kawato, 1998*). This architecture allowed us to track the value function of each RL expert separately, while using a unique, global action in each trial.

Estimated hyperparameters (learning rate α, forgetting factor γ, RPE variance ν) were used to compute value functions of participant data, as well as to generate new, artificial choice data and value functions (*Figure 2C*, and Materials and methods). Simulations indicated that expected value and responsibility were highest for the appropriate Abstract RL, followed by Feature RL, and the two Abstract RLs based on irrelevant features as the lowest (*Figure 2D*). Participant empirical data displayed the same pattern, whereby the value function and responsibility signal of the appropriate Abstract RL were higher than in other RL algorithms (*Figure 2D*, right side). Note that the large difference between appropriate Abstract RL and Feature RL was because the appropriate Abstract RL was an 'oracle': it had access to the correct low-dimensional state from the beginning. The RPE variance (hyperparameter ν) adjusted the sharpness with which each RL's (un)certainty was considered for expert weighting. Crucially, the variance **ν** was associated with participant learning speed, such that participants who learned block rules quickly were sharper in expert selection (*Figure 2E*, N = 29, robust regression slope = −1.02, $t_{27}$ = −2.59, p = 0.015). These modelling results provided a first layer of support for the hypothesis that valuation is related to abstraction.

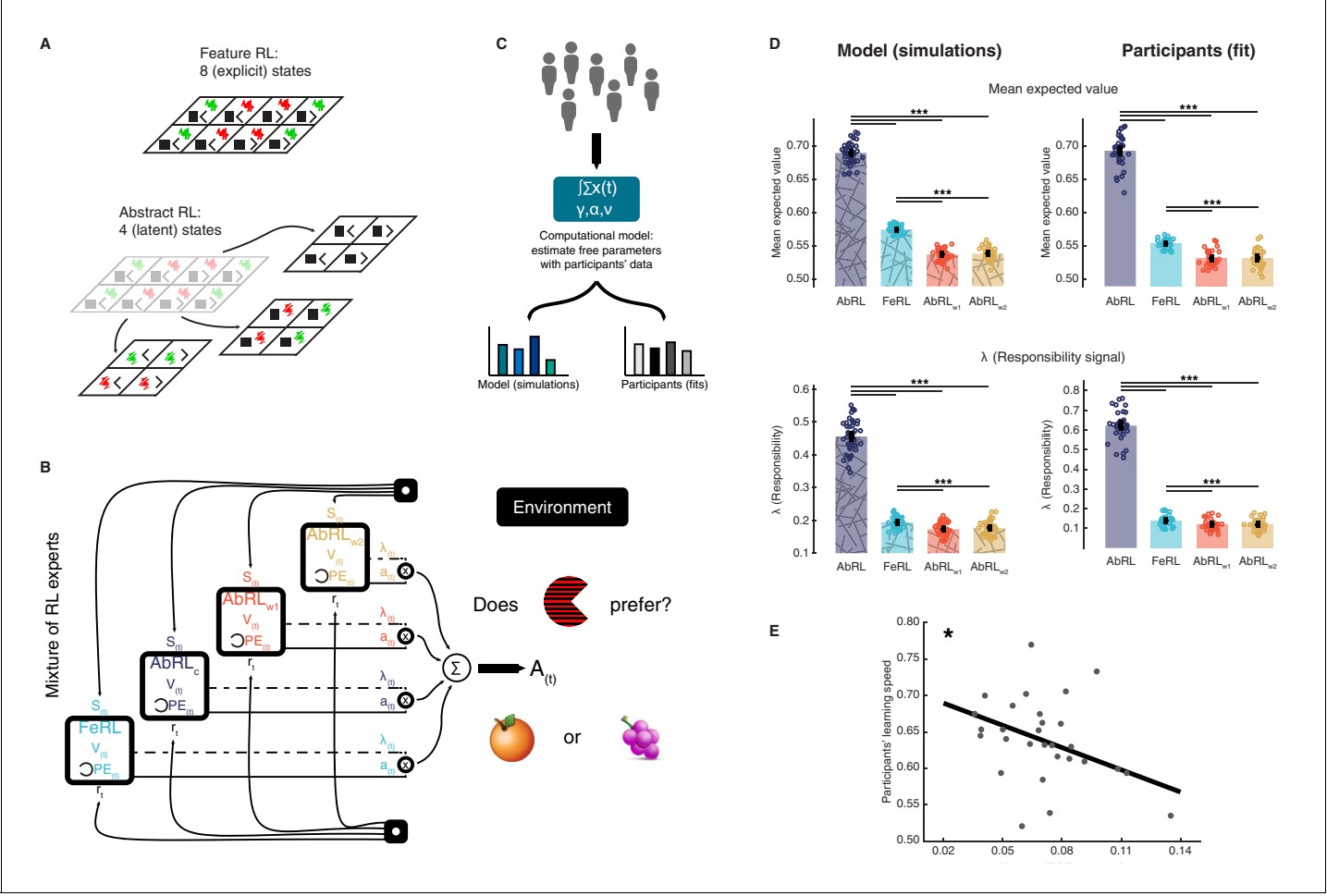

**Figure 2.** Mixture of reinforcement learning (RL) experts and value computation. (A) Outline of the representational spaces of each RL algorithm comprising the mixture-of-experts architecture. (B) Illustration of the model architecture. See Methods for a formal description of the model. All experts had the same number of hyperparameters: the learning rate $\alpha$ (how much the latest outcome affected agent beliefs), the forgetting factor $\gamma$ (how much prior RPEs influenced current decisions), and the RPE variance $\nu$, modulating the sharpness with which the mixture-of-expert RL model should favour the best performing algorithm in the current trial. (C) The approach used for data analysis and model simulation. The model was first fitted to participant data with Hierarchical Bayesian Inference (*Piray et al., 2019*). Estimated hyperparameters were used to compute value functions of participant data, as well as to generate new, artificial choice data and to compute simulated value functions. (D) Averaged expected value across all states for the chosen action in each RL expert, as well as responsibility signal for each model. Left: simulated data, right: participant empirical data. Dots represent individual agents (left) or participants (right). Bars indicate the mean and error bars depict the SEM. Statistical comparisons were performed with two-sided Wilcoxon signed rank tests. ***$p<0.001$. AbRL: Abstract RL, FeRL: Feature RL, AbRL$_{w1}$: wrong-1 Abstract RL, AbRL$_{w2}$: wrong-2 Abstract RL. (E) RPE variance was negatively correlated with learning speed (outliers removed, N = 29). Dots represent individual participant data. The thick line shows the linear regression fit. The experiment was conducted once (n = 33 biologically independent samples), * $p<0.05$.

The online version of this article includes the following source data and figure supplement(s) for figure 2:

Source data 1. Csv: panel D, mean expected value, model.
Source data 2. Csv: panel D, mean expected value, subjects.
Source data 3. Csv: panel D, lambda, model.
Source data 4. Csv: panel D, lambda, subjects.
Source data 5. Csv: panel E.
Figure supplement 1. Model comparison (accounting for model complexity).

## Behaviour shifts from Feature- to Abstraction-based reinforcement learning

The mixture-of-expert RL model revealed that participants who learned faster relied more on the best RL model value representations. Further, the modelling established that choices were mostly

driven by either the appropriate Abstract RL or Feature RL, which had higher expected values (but note that the other Abstract RLs had mean values greater than a null level of 0.5), and higher responsibility λ. It is important to highlight though that the mixture-of-experts RL might not reflect the actual algorithmic computation used by the participant in this task, but it provides a conceptual solution to the arbitration between representations/strategies. Model comparison showed that Abstract RL and Feature RL in many cases offered a more parsimonious description of the participants behaviour. This is unsurprising since Feature RL is a simple model and Abstract RL is an oracle model – knowing which are the relevant feature (see *Figure 2—figure supplement 1* for a direct model comparison between mixture-of-experts RL, Feature RL, and Abstract RL). Hence, we next sought to explicitly explain participant choices and learning according to either Feature RL or Abstract RL strategy. Given the task space (*Figure 2A*), the only way to solve a block rule faster was to use abstract representations. As such, we expect to observe a shift from Feature RL toward Abstract RL to occur with learning.

Both algorithms had two hyperparameters: the learning rate α and greediness β (inverse temperature, the strength that expected value has on determining actions). Using the estimated hyperparameters, we generated new, synthetic data to evaluate how fast artificial agents, implementing either Feature RL or Abstract RL, solved the learning task (see Materials and methods). The simulations attested that Feature RL was slower and less efficient (*Figure 3A*), yielding lower learning speed and a higher percentage of failed blocks.

Model comparison at the single participant and block levels (*Piray et al., 2019*) provided a direct way to infer which algorithm was more likely to explain learning in any given block. Overall, similar proportions of blocks were classified as Feature RL and Abstract RL. This indicates that participants used both learning strategies (binomial test applied to all blocks: proportion of Abstract RL = 0.47 vs. equal level = 0.5, $P(212|449) = 0.26$, *Figure 3B*; two-sided t-test of participant-level proportions: lower, but close to 0.5, $t_{32} = -2.87$, p = 0.007, *Figure 3B* inset).

As suggested by the simulations (*Figure 3A*), the strategy that best explained participant block data accounted for the distribution of learning speed measures in each block. Where learning proceeded slowly, Feature RL was consistently predominant (*Figure 3B*), while the reverse happened in blocks where participants displayed fast learning (*Figure 3B*). Among participants, the degree of abstraction (propensity to use Abstract RL) correlated with the empirical learning speed (N = 33, robust regression, slope = 0.52, $t_{31} = 4.56$, p = 7.64x$10^{-5}$, *Figure 3C* top). Participant confidence in having performed the task well was also significantly correlated with the degree of abstraction (N = 31, robust regression, slope = 0.026, $t_{29} = 2.69$, p = 0.012, *Figure 3C*, bottom). In addition to the finding that confidence related to learning speed (*Figure 1E*), these results raise intriguing questions about the function of metacognition, as participants appeared to comprehend their own ability to construct and use abstractions (*Cortese et al., 2020*).

The two RL algorithms revealed a second aspect of learning. Considering all blocks regardless of fit (paired comparison), feature RL appeared to have higher learning rates α compared with Abstract RL (two-sided Wilcoxon rank sum test against median 0, z = 14.33, p < $10^{-30}$, *Figure 3D*). A similar asymmetry was found with greediness (*Figure 3E*, two-sided Wilcoxon rank sum test against median 0, z = 7.14, p < $10^{-10}$). Yet, more specifically, considering only the model (Feature RL or Abstract RL) which provided the best fit on a given block, resulted in Feature RL displaying *lower* learning rates and greediness (*Figure 3—figure supplement 1A*). The order inverted entirely when considering the model which provided the *worst* fit: higher learning rates and greediness for Feature RL (*Figure 3—figure supplement 1B*). These differences can be explained intuitively as follows. In Feature RL, exploration of the task state-space takes longer - in short blocks (best fit by the Abstract RL strategy) a higher learning rate is necessary for the Feature RL agent to make larger updates on states that are infrequently visited. Results also suggest that action selection tends to follow the same principles – more deterministic in blocks that are best fit by Abstract RL (i.e. large β for shorter blocks).

We predicted that use of abstraction should increase with learning, because the brain has to construct abstractions in the first place and must initially rely on Feature RL. To test this hypothesis, we quantified the number of participants using a Feature RL or Abstract RL strategy in their first and last blocks. On their first block, most participants relied on Feature RL, while the pattern reversed in the last block (two-sided sign test, z = −2.77, p = 0.006, *Figure 3F*). Computing the abstraction level separately for the session median split of early and late blocks also resulted in higher abstraction in late blocks (two-sided sign test, z = −2.94, p = 0.003, *Figure 3G*). These effects were

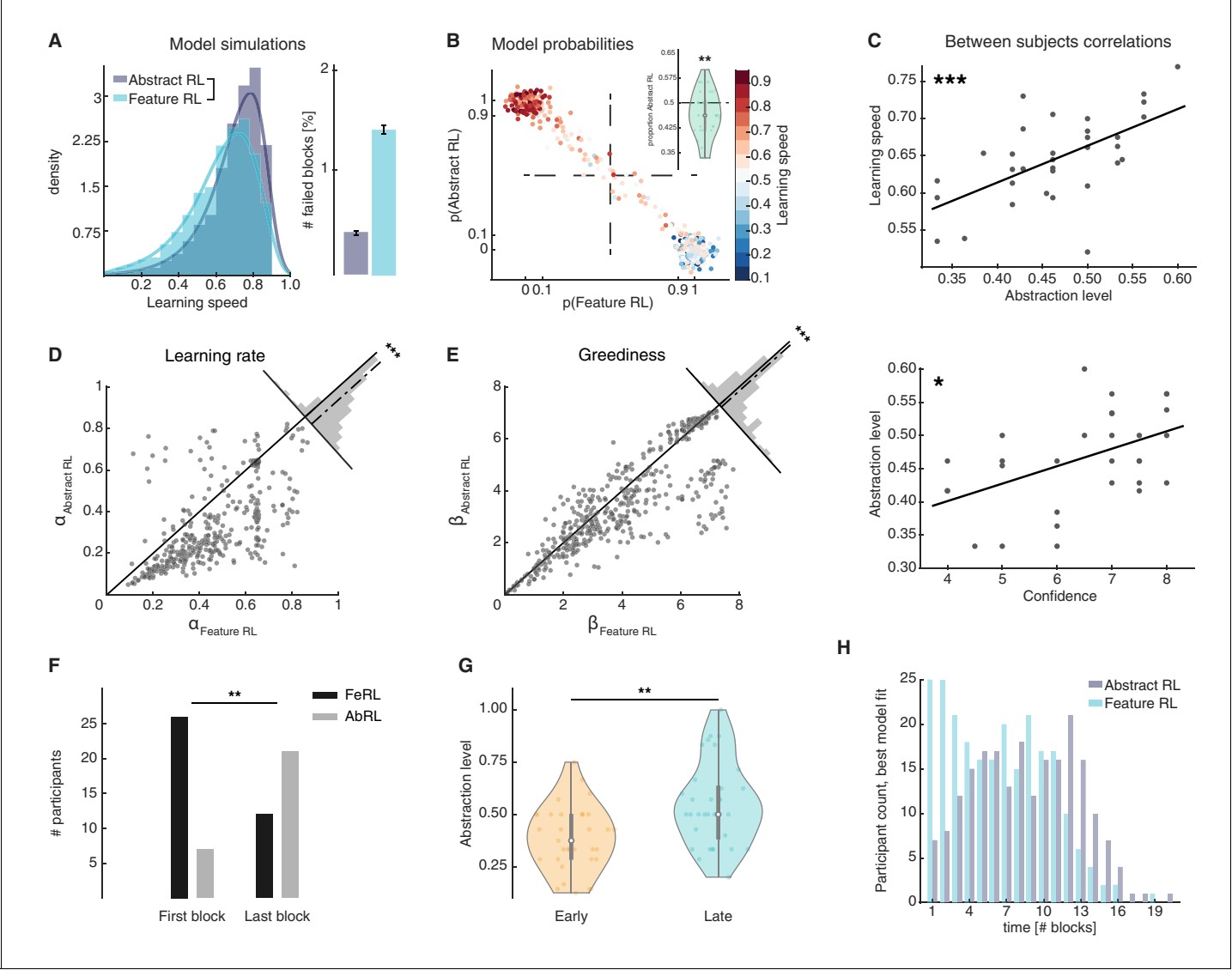

**Figure 3.** Feature RL vs Abstract RL are related to learning speed and the use of abstraction increases with experience. (A) Simulated learning speed and % of failed blocks for both Abstract RL and Feature RL. To make simulations more realistic, arbitrary noise was injected into the simulation, altering the state (see Materials and methods). N = 100 simulations of 45 agents. Right plot: bars represent the mean, error bars the SEM. (B) The relationship between the block-by-block, best-fitting model and learning speed of participants. Each dot represents one block from one participant, with data aggregated across all participants. Note that some dots fall beyond p=one or p=0. This effect occurs because dots were scattered with noise in their x-y coordinates for better visualisation. (C) Between participant correlations. Top: abstraction level vs learning speed. The abstraction level was computed as the average over all blocks completed by a given participant (code: Feature RL = 0, Abstract RL = 1). Bottom: confidence vs abstraction level. Dots represent individual participants (top: N = 33, bottom: N = 31, some dots are overlapping). (D) Learning rate was not symmetrically distributed across the two algorithms. (E) Greediness was not symmetrically distributed across the two algorithms. For both (D and E), each dot represents one block from one participant, with data aggregated across all participants. Histograms represent the distribution of data around the midline. (F) The number of participants for which Feature RL or Abstract RL best explained their choice behaviour in the first and last blocks of the experimental session. (G) Abstraction level was computed separately with blocks from the first half (early) and latter half (late) session. (H) Participants count for the best fitting model, in each block. The experiment was conducted once (n = 33 biologically independent samples), * p<0.05, ** p<0.01, *** p<0.001.

The online version of this article includes the following source data and figure supplement(s) for figure 3:

**Source data 1.** Csv: panel A left, model simulations histogram of learning speed.
**Source data 2.** Csv: panel A right, model simulations % failed blocks.
**Source data 3.** Csv: panel B, scatter plot of model probabilities.
**Source data 4.** Csv: panel B, violin plot of proportion Abstract RL.
**Source data 5.** Csv: panel C.

*Figure 3 continued on next page*

*Figure 3 continued*

**Source data 6.** Csv: panel D.
**Source data 7.** Csv: panel E.
**Source data 8.** Csv: panels F and G.
**Source data 9.** csv: panel H.
**Figure supplement 1.** Comparison of learning rate α and greediness β in Feature RL and Abstract RL best-fitting and worst-fitting blocks.
**Figure supplement 2.** Abstraction index for single blocks and expected value for the chosen action in Abstract RL and Feature RL.
**Figure supplement 3.** Parameter recovery.
**Figure supplement 4.** Strategy analysis of excluded participants.

complemented by two block-by-block analyses, displaying an increase in abstraction from early to late blocks (*Figure 3H*, and *Figure 3—figure supplement 2A*).

Supporting the current modelling framework, the mean expected value of the chosen action was higher for Abstract RL (*Figure 3—figure supplement 2B–C*), and model hyperparameters could be recovered in the presence of noise (*Figure 3—figure supplement 3*; see Materials and methods) (*Palminteri et al., 2017*). Given the lower learning speed in excluded participants, the distribution of strategies was also different among them, with a higher ratio of Feature RL blocks (*Figure 3—figure supplement 4*).

## The role of vmPFC in constructing goal-dependent value from sensory features

The computational approach confirmed that participants relied on both a low-level feature strategy, and a more sophisticated abstract strategy (i.e. Feature RL and Abstract RL; *Figures 2D* and *3B*). Beside proving that abstract representations were generally associated with higher expected value, the modelling approach further allowed us to explicitly *classify* trials as belonging to either learning strategy. Here, our goal was to dissociate neural signatures of these distinct learning strategies in order to show how abstract representations are constructed by the human brain.

First, we reasoned that an anticipatory value signal might emerge in the vmPFC at stimulus presentation (*Knutson et al., 2005*). We performed a general linear model (GLM) analysis of neuroimaging data with regressors for 'High-value' and 'Low-value' trials, selected by the block-level best fitting algorithm (Feature RL or Abstract RL, while controlling for other confounding factors such as time and strategy itself; see Materias and methods and Supplementary note one for the full GLM and regressors specification). As predicted, activity in the vmPFC strongly correlated with value magnitude (*Figure 4A*). That is, the vmPFC indexed the anticipated value constructed from pacman features at stimulus presentation time. We used this signal to functionally define, for ensuing analyses, the subregion of the vmPFC that was maximally related to task computations about value when pacman visual features were integrated. Concurrently, activity in insular and dorsal prefrontal cortices increased under conditions of low expected value. This pattern of activity is consistent with previous studies on error monitoring and processing (*Bastin et al., 2017*; *Carter et al., 1998*; *Figure 4—figure supplement 1*).

In order for the vmPFC to construct goal-dependent value signals, it should receive relevant feature information from other brain areas and specifically from visual cortices, given the nature of our task. Thus, we computed a psychophysiological interaction (PPI) analysis (*Friston et al., 1997*), to isolate regions in which functional coupling with the vmPFC at the time of stimulus presentation was dependent on the magnitude of expected value. Supporting the idea that the vmPFC based its predictions on the integration of visual features, only connectivity between the visual cortex (VC) and vmPFC was higher on trials that carried large expected value, compared to low-value trials (*Figure 4B*). Strikingly, the strength of this VC - vmPFC interaction was associated with the overall learning speed among participants (N = 31, robust regression, slope = 0.016, $t_{29}$ = 2.55, p = 0.016, *Figure 4C*), such that participants with stronger modulation of the coupling between the vmPFC and VC also learned block rules faster. The strength of the vmPFC - VC coupling showed a non-significant trend with the level of abstraction (N = 31, robust regression, slope = 0.013, $t_{29}$ = 1.56, p = 0.065 one-sided, *Figure 4—figure supplement 2*). However, this study was not optimised to detect between subject correlations that normally require a larger number of subjects. Therefore, future work is required to confirm or falsify this result.

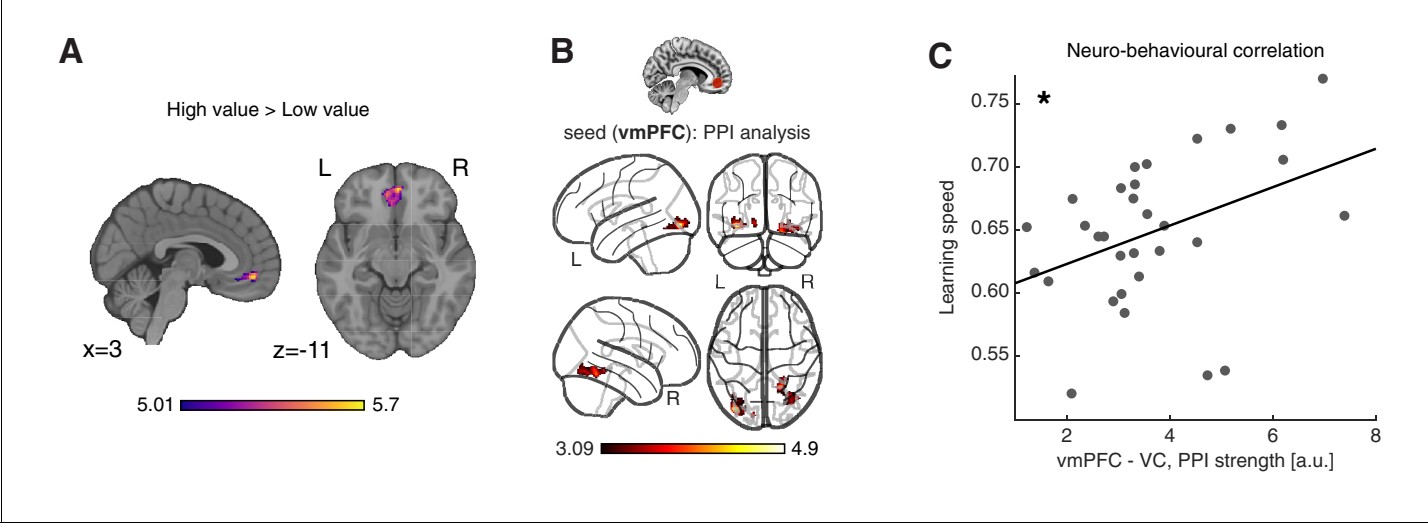

**Figure 4.** Neural substrates of value construction during learning. (**A**) Correlates of anticipated value at pacman stimulus presentation time. Trials were labelled according to a median split of the expected value for the chosen action, as computed by the best fitting model, Feature RL or Abstract RL, at the block level. Mass univariate analysis, contrast 'High-value' > 'Low-value'. vmPFC peaks at [2 50 -10]. The statistical parametric map was z-transformed and plotted at $p$(FWE) < 0.05. (**B**) Psychophysiological interaction, using as seed a sphere (radius = 6 mm) centred around the participant-specific peak voxel, constrained within a 25 mm sphere centred around the group-level peak coordinate from contrast in (**A**). The statistical parametric map was z-transformed and plotted at $p$(fpr) < 0.001 (one-sided, for positive contrast - increased coupling). (**C**) The strength of the interaction between the vmPFC and VC was positively correlated with participant's ability to learn block rules. Dots represent individual participant data points, and the line is the regression fit. The experiment was conducted once (n = 33 biologically independent samples), * p<0.05.

The online version of this article includes the following source data and figure supplement(s) for figure 4:

**Source data 1.** Csv: panel C.
**Figure supplement 1.** 'Low value' > 'High value' GLM contrast.
**Figure supplement 2.** Neuro-behavioural correlation between VC-vmPFC coupling and abstraction.

## A value-sensitive vmPFC subregion prioritises abstract elements

Having established that the vmPFC computes a goal-dependent value signal, we evaluated whether the activity level of this region was sensitive to the strategies that participants used. To do so, we used the same GLM introduced earlier, and estimated two new statistical maps from the regressors 'Abstract RL' and 'Feature RL', while controlling for idiosyncratic features of the task, that is, high/low value and early/late trials (see Materials and methods and Supplementary note 1). We extracted the peak activity at the participant level, under Feature RL and Abstract RL conditions, in two regions-of-interest (ROI). Specifically, we focused on the vmPFC and the HPC, as both have been consistently linked with abstraction, and feature-based and conceptual learning. The HPC was defined anatomically (AAL atlas, *Figure 5A* top), while the vmPFC was defined as voxels sensitive to the orthogonal contrast 'High value' > 'Low value' from the same GLM (*Figure 5A* bottom). A linear mixed effects model (LMEM) with fixed effects 'ROI' and 'strategy' [LMEM: 'y ~ ROI * strategy + (1| participants)', y: ROI activity] revealed significant main effects of 'ROI' ($t_{128}$ = 2.16, $p$ = 0.033), and 'strategy' ($t_{128}$ = 3.07, p = 0.003), and a significant interaction ($t_{128}$ = −2.29, p = 0.024), illustrating different HPC and vmPFC recruitment (*Figure 5B*). Post-hoc comparisons showed vmPFC activity levels distinguished Feature RL and Abstract RL cases well (LMEM: $t_{64}$ = 2.94, $p_{(FDR)}$ = 0.009), while the HPC remained agnostic to the style of learning (LMEM: $t_{64}$ = 0.62, $p_{(FDR)}$ = 0.54). Alternative explanations are unlikely, as there was no effect in terms of both the correlation between value-type trials and algorithms, and task difficulty, measured by reaction times (*Figure 5—figure supplement 2*).

The next question we asked was, 'Can we retrieve feature information from HPC and vmPFC activity patterns?' In order to abstract and operate in the latent space, an agent is still bound to represent and use the features, because the rules are dictated by feature combinations. One possibility is that feature information is represented solely in sensory areas. What matters then is the connection with and/or the read out of vmPFC or HPC. Accordingly, neither HPC nor vmPFC should

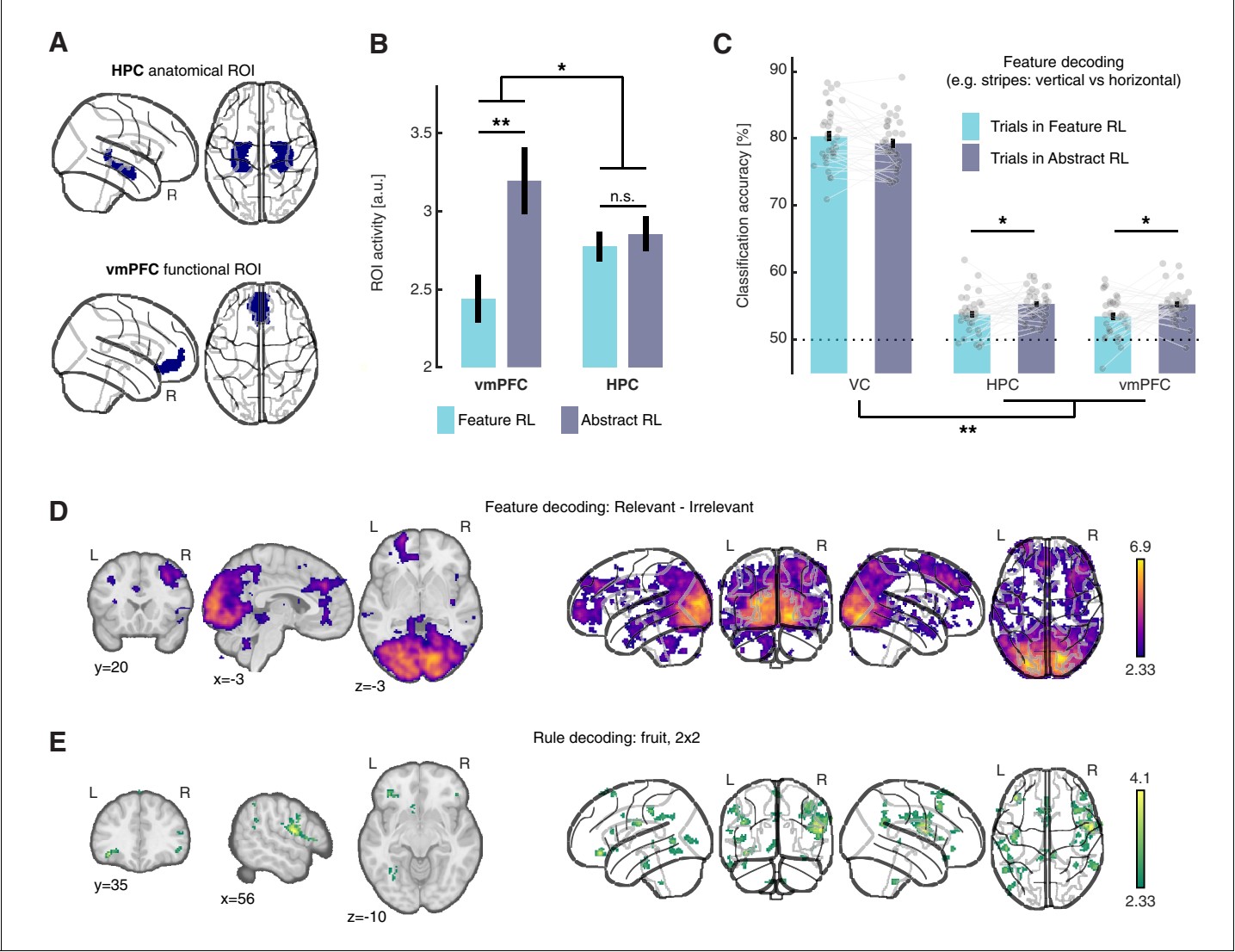

**Figure 5.** Neural substrate of abstraction. (**A**) Regions of interest for univariate and multivariate analyses. The HPC was defined through automated anatomical labelling (FreeSurfer). The vmPFC was functionally defined as the cluster of voxels found with the orthogonal contrast 'High value' > 'Low value', at $P$(unc) < 0.0001. (**B**) ROI activity levels corresponding to each learning mode were extracted from the contrasts 'Feature RL' > 'Abstract RL', and 'Abstract RL' > 'Feature RL'. Coloured bars represent the mean, and error bars the SEM. (**C**) Multivariate (decoding) analysis in three regions of interest: VC, HPC, vmPFC. Binary decoding was performed for each feature (e.g. colour: red vs green), by using trials from blocks labelled as Feature RL or Abstract RL. Colour bars represent the mean, error bars the SEM, and grey dots represent individual data points (for each individual, taken as the average across all three classifications, i.e., of all features). Results were obtained from leave-one-run-out cross-validation. The experiment was conducted once (n = 33 biologically independent samples), * p<0.05, ** p<0.01. (**D**) Classification was performed for each feature pair (e.g. colour: red vs green), separately for blocks in which the feature in question was relevant or irrelevant to the block's rule. The statistical map represents the strength of the reduction in accuracy between trials in which the feature was relevant compared to irrelevant, averaged over all features and participants. (**E**) Classification of the rule (2x2 blocks only). For each participant, classification was performed as fruit 1 vs fruit 2. In (**D–E**), statistical parametric maps were z-transformed, false-positive means of cluster formation (fpr) correction was applied. p(fpr) < 0.01, Z > 2.33.

The online version of this article includes the following source data and figure supplement(s) for figure 5:

**Source data 1.** Csv: panel B.
**Source data 2.** Csv: panel C.
**Figure supplement 1.** Ratio correct in Feature RL and Abstract RL.
**Figure supplement 2.** Value functions correlations and reaction time differences in Feature RL and Abstract RL trials.
**Figure supplement 3.** K-fold cross-validation in feature decoding, using Feature RL and Abstract RL trials separately.

represent feature information, regardless of the strategy used. Alternatively, feature-level information could also be represented in higher cortical regions under Abstract RL to explicitly support (value-based) relational computations (*Oemisch et al., 2019*). To resolve this question, we applied multivoxel pattern analysis to classify basic feature information (e.g. colour: red vs green) in three regions of interest: the VC, HPC, and vmPFC, separately for trials labelled as Feature RL or Abstract RL. We found that classification accuracy was significantly higher in Abstract RL trials compared with Feature RL trials in both the HPC and vmPFC (two-sided t-test, HPC: $t_{32}$ = −2.37, $p_{(FDR)}$ < 0.036, vmPFC: $t_{32}$ = −2.51, $p_{(FDR)}$ = 0.036, *Figure 5C*), while the difference was of opposite sign in VC ($t_{32}$ = 1.61, $p_{(FDR)}$ = 0.12, *Figure 5C*). The increased feature decodability in Abstract RL was significantly larger in the HPC and vmPFC compared to the VC (LMEM model 'y ~ ROI + (1|participants)', y: difference in decodability, $t_{97}$ = 3.37, p = 0.001). Due to the nature of the task, the number of trials in each category could vary and thus confounds the analysis. A control analysis equating the number of training trials for each feature and condition replicated the original finding (*Figure 5—figure supplement 3*). These empirical results support the second hypothesis. In Abstract RL, features are represented in the neural circuitry incorporating the HPC and vmPFC, beyond a simple read out of sensory cortices. In Feature RL, representing feature-level information in sensory cortices alone should suffice because each visual pattern mapped to a task-state.

We expanded on this idea with two searchlight multivoxel pattern analyses. In short, we inquired which brain regions are sensitive to feature relevance, and whether we could recover representations of the latent rule itself (the fruit preference). Beside the occipital cortex, significant reduction in decoding accuracy was also detected in the OFC, ACC, vmPFC and dorsolateral PFC when a feature was irrelevant to the rule, compared to when it was relevant (*Figure 5D*). Multivoxel patterns in the dorsolateral PFC and lateral OFC further predicted fruit class (*Figure 5E*).

## Artificially injecting value in sensory representations with neurofeedback fosters abstraction

Our computational and neuroimaging results indicate that valuation serves a key function in abstraction. Two hypotheses on the underlying mechanism can be outlined here. On one hand, the effect of vmPFC value computations could remain localised within the prefrontal circuitry. For example, this could be achieved by representing and ranking incoming sensory information for further processing within the HPC-OFC circuitry. Alternatively, value computation could determine abstractions by directly affecting early sensory areas – that is, a top-down (attentional) effect to 'tag' relevant sensory information (*Anderson et al., 2011*). Work in rodents has reported strong top-down modulation of sensory cortices by OFC neurons implicated in value computations (*Banerjee et al., 2020*; *Liu et al., 2020*). We thus hypothesised that abstraction could result from a direct effect of value in the VC. Therefore, artificially adding value to a neural representation of a task-relevant feature should result in enhanced behavioural abstraction.

Decoded neurofeedback is a form of neural reinforcement based on real time fMRI and multivoxel pattern analysis. It is the closest approximation to a non-invasive causal manipulation, with high specificity and administered without participant awareness (*Lubianiker et al., 2019*; *Muñoz-Moldes and Cleeremans, 2020*; *Shibata et al., 2019*). Such reinforcement protocols can reliably lead to persistent behavioural or physiological changes (*Cortese et al., 2016*; *Koizumi et al., 2016*; *Shibata et al., 2011*; *Sitaram et al., 2017*; *Taschereau-Dumouchel et al., 2018*). We used this procedure in a follow-up experiment (N = 22, a subgroup from the main experiment; see Materials and methods) to artificially add value (rewards) to neural representation in VC of a target task-related feature (*Figure 6A*). At the end of two training sessions, participants completed 16 blocks of the pacman fruit preference task, outside of the scanner. Task blocks could be labelled as 'relevant' (eight blocks) if the feature tagged with value was relevant to the block rule, or 'irrelevant' otherwise (eight blocks).

Data from the 'relevant' and 'irrelevant' blocks were analysed separately. The same model-fitting procedure used in the main experiment established whether participant choices in the new blocks were driven by a Feature RL or Abstract RL strategy. 'Relevant' blocks appeared to be associated with a behavioural shift toward Abstract RL, whereas there was no substantial qualitative effect in 'irrelevant' blocks (*Figure 6B*). To quantify this effect, we first applied a binomial test, finding that behaviour in blocks where the targeted feature was relevant displayed markedly increased abstraction (base rate 0.5, number of Abstract RL blocks given total number of blocks; 'relevant': P(123|176)

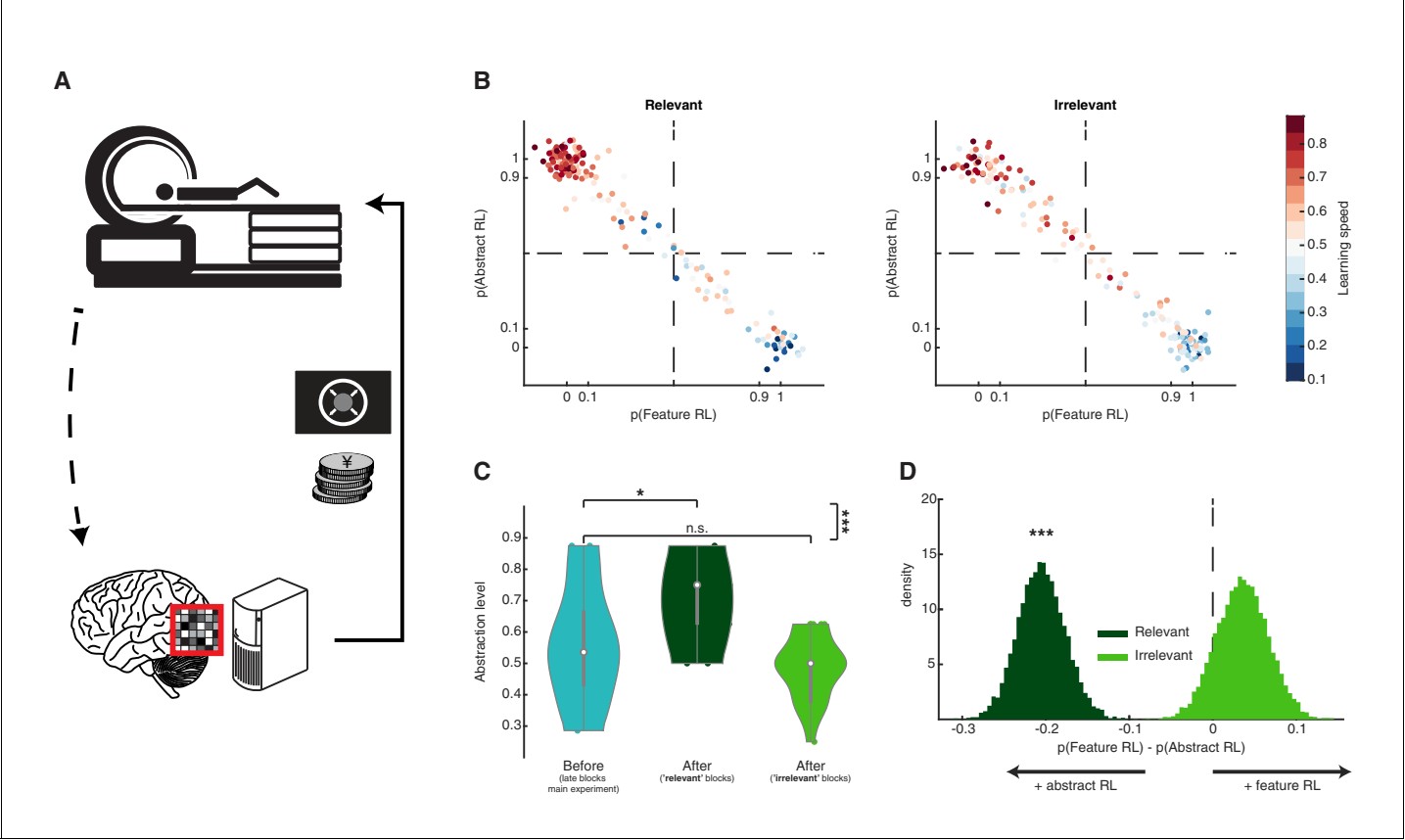

**Figure 6.** Artificially adding value to a feature's neural representation. (**A**) Schematic diagram of the follow-up multivoxel neurofeedback experiment. During the neurofeedback procedure, participants were rewarded for increasing the size of a disc on the screen (max session reward 3000 JPY). Unbeknownst to them, disc size was changed by the computer program to reflect the likelihood of the target brain activity pattern (corresponding to one of the task features) measured in real time. (**B**) Blocks were subdivided based on the feature targeted by multivoxel neurofeedback as 'relevant' or 'irrelevant' to the block rules. Scatter plots replicate the finding from the main experiment, with a strong association between Feature RL / Abstract RL and learning speed. Each coloured dot represents a single block from one participant, with data aggregated from all participants. (**C**) Abstraction level was computed for each participant from all blocks belonging to: (1) left, the latter half of the main experiment (as in *Figure 3G*, but only selecting participants who took part in the multivoxel neurofeedback experiment); (2) centre, post-neurofeedback for the 'relevant' condition; (3) right, post-neurofeedback for the 'irrelevant' condition. Coloured dots represent participants. Shaded areas indicate the density plot. Central white dots show the medians. The dark central bar depicts the interquartile range, and dark vertical lines indicate the lower and upper adjacent values. (**D**) Bootstrapping the difference between model probabilities on each block, separately for 'relevant' and 'irrelevant' conditions. The experiment was conducted once (n = 22 biologically independent samples), * p<0.05, *** p<0.001.

The online version of this article includes the following source data and figure supplement(s) for figure 6:

**Source data 1.** Csv: panel B, irrelevant blocks.
**Source data 2.** Csv: panel B, relevant blocks.
**Source data 3.** Csv: panel C.
**Source data 4.** Csv: panel D.
**Figure supplement 1.** Neurofeedback experiment results.

= $1.37 \times 10^{-7}$, 'irrelevant': P(90|176) = 0.82). We then measured the abstraction level for each participant and directly compared it to the level attained by the same participants in the late blocks of main experiments (from *Figure 3G*). Participants increased their use of abstraction in 'relevant' blocks, whereas no significant difference was detected in the 'irrelevant' blocks (*Figure 6C*, two-sided Wilcoxon signed rank test, 'relevant' blocks: z = 2.44, p = 0.015, 'irrelevant' blocks: z = −1.55, p = 0.12, 'relevant' vs 'irrelevant': z = 4.01, p = $6.03 \times 10^{-5}$). Finally, we measured the difference between model probabilities $P_{(Feature\ RL)}$ - $P_{(Abstract\ RL)}$ for each block, and bootstrapped the mean over blocks (with replacement) 10,000 times to generate a distribution for 'relevant' and 'irrelevant' conditions. Replicating the results reported above, behaviour in 'relevant' blocks was more likely to

be driven by Abstract RL (*Figure 6D*, perm. test p < 0.001), while Feature RL tended to appear more in 'irrelevant' blocks. Participants were successful at increasing the disk size in the neurofeedback task (*Figure 6—figure supplement 1A–B*). Furthermore, those who were more successful were also more likely to display larger increases in abstraction in the subsequent behavioural test compared to their initial level (*Figure 6—figure supplement 1C*).

A strategy shift toward abstraction, specific to blocks in which the target feature was tagged with reward, indicates that the effect of value in facilitating abstraction is likely to be mediated by a change in the early processing stage of visual information. In this experiment, by means of neurofeedback, value (in the form of external rewards) 'primed' a target feature. Hence, the brain used these 'artificial' values when constructing abstract representations by tagging certain sensory channels. Critically, this manipulation indicates that value tagging of early representation has a causal effect on abstraction and consequently on the learning strategy.

## Discussion

The ability to generate abstractions from simple sensory information has been suggested as crucial to support flexible and adaptive behaviours (*Cortese et al., 2019*; *Ho et al., 2019*; *Wikenheiser and Schoenbaum, 2016*). Here, using computational, we found that value predictions drive participant selections of the appropriate representation to solve the task. Participants explored and used task dimensionality through learning, as they shifted from a simple feature-based strategy to using more sophisticated abstractions. The more participants used Abstract RL, the faster they became at solving the task. Note that in this task, structure, learning speed, and abstraction are linked. To learn faster, an agent must use Abstract RL, as other strategies would result in slower completion of task blocks.

These results build on the idea that efficient decision-making processes in the brain depend on higher-order, summarised representations of task-states (*Niv, 2019*; *Schuck et al., 2016*). Further, abstraction likely requires a functional link between sensory regions and areas encoding value predictions about task states (*Figure 4C*, the VC-vmPFC coupling was positively correlated with participant's learning speed). This is consistent with previous work that demonstrates how estimating reward value of individual features provides a reliable and adaptive mechanism in RL (*Farashahi et al., 2017*). We extend this notion by showing that the mechanism may support formation of abstract representations to be further used for learning computations, for example selection of the appropriate abstract representation.

An interesting question concerns whether the brain uses abstract representations in isolation - operating in a hypothesis-testing regime - that is, favouring the current best model; or whether representations may be used to update multiple internal models, with behaviour determined by their synthesis (as in the mixture-of-experts architecture). The latter implementation may not be the most efficient computationally - the brain would have to run multiple processes in parallel, but it would be very data efficient since one data point can be used to update several models. Humans might (at least at the conscious level) engage with one hypothesis at the time. However, there is circumstantial evidence that multiple strategies might be computed in parallel but deployed one at the time (*Domenech et al., 2018*; *Donoso et al., 2014*). Furthermore, in many cases, the brain may have access to only limited data points, while parallel processing is a major feature of neural circuits (*Alexander and Crutcher, 1990*; *Lee et al., 2020*; *Spitmaan et al., 2020*). In this study, we aimed to show that arbitration between feature and abstract learning may be achieved using a relatively simple algorithm (the mixture-of-experts RL) and then proceeded to characterise the neural underpinnings of these two types of learning (i.e. Feature RL and Abstract RL). Admittedly, in the present work the mixture-of-experts RL does not provide a solid account of the data when compared to the more parsimonious Feature RL and Abstract RL in isolation. Future work will need to establish the actual computational strategy employed by the human brain. Of particular importance will be further examining how such strategies vary across circumstances (tasks, contexts, or goals).

There is an important body of work considering how the HPC is involved in formation and update of conceptual information (*Bowman and Zeithamova, 2018*; *Kumaran et al., 2009*; *Mack et al., 2016*; *McKenzie et al., 2014*). Likely, the role of the HPC is to store, index, and update conceptual/schematic memories (*Mack et al., 2016*; *Tse et al., 2011*; *Tse et al., 2007*). 'Creation' of new concepts or schemas may occur elsewhere. A good candidate could be the mPFC or the vmPFC in

humans (*Mack et al., 2020*; *Tse et al., 2011*). Indeed, the vmPFC exhibits value signals directly modulated by cognitive requirements (*Castegnetti et al., 2021*). Our results expose a potential mechanism of how the vmPFC interacts with the HPC in construction of goal-relevant abstractions. vmPFC-driven valuation of low-level sensory information serves to channel-specific features/components to higher order areas (e.g. the HPC, vmPFC, but also the dorsal prefrontal cortex, for instance). Congruent with this interpretation, we found that the vmPFC was more engaged in Abstract RL, while the HPC was equally active under both abstract and feature-based strategies (*Figure 5B*). When a feature was irrelevant to the rule, its decodability from activity patterns in OFC/DLPFC decreased (*Figure 5D*). These findings accord well with the role of prefrontal regions in constructing goal-dependent task states and abstract rules from relevant sensory information (*Akaishi et al., 2016*; *Schuck et al., 2016*; *Wallis et al., 2001*). Furthermore, we found block rules were also encoded in multivoxel activity patterns within the OFC/DLPFC circuitry (*Figure 5E*; *Bengtsson et al., 2009*; *Mian et al., 2014*; *Wallis et al., 2001*).

How these representations are actually used remains an open question. This study nevertheless suggests that there is no single region of the brain that maintains a fixed task state. Rather, the configuration of elements that determines a state is continuously reconstructed over time. At first glance, this may appear costly and inefficient. But this approach would provide high flexibility in noisy and stochastic environments, and where temporal dependencies exist (as in most real-world situations). By continuously recomputing task states, the agent can make more robust decisions because these are related to the subset of most relevant, up-to-date information. Such a computational coding scheme shares strong analogies with HPC neural coding, whereby neurons continuously generate representations of alternative future trajectories (*Kay et al., 2020*), and replay past cognitive trajectories (*Schuck and Niv, 2019*).

One significant topic for discussion concerns the elements used to construct abstractions. We leveraged simple visual features (colour, or stripe orientation), rather than more complex stimuli or objects that can be linked conceptually (*Kumaran et al., 2009*; *Zeithamova et al., 2019*). Abstractions happen at several levels, from features, to exemplars, concepts/categories, and all the way to rules and symbolic representations. In this work, we effectively studied one of the lowest levels of abstraction. One may wonder whether the mechanism we identified here generalises to more complex scenarios. While our work cannot decisively support this, we believe it unlikely that the brain uses an entirely different strategy to generate new representations at different levels of abstraction. Rather, the internal source of information abstracted should be different, but the algorithm itself should be the same or, at the very least, highly similar. The fact our PPI analysis showed a link between the vmPFC and VC may point to this distinguishing characteristic of our study. Learning through abstraction of simple visual features should be related to early VC. Features in other modalities, for example, auditory, would involve functional coupling between the auditory cortex and the vmPFC. When learning about more complex objects or categories, we expect to see stronger reliance on the HPC (*Kumaran et al., 2009*; *Mack et al., 2016*). Future studies could incorporate different levels of complexity, or different modalities, within a similar design so as to directly test this prediction and dissect exact neural contributions. Depending on which type of information is relevant at any point in time, we suspect that different areas will be coupled with the vmPFC to generate value representations.

In our second experiment, we implemented a direct assay to test our (causal) hypothesis that valuation of features guides abstraction in learning. Artificially adding value in the form of reward to a feature representation in the VC resulted in increased use of abstractions. Thus, the facilitating effect of value on abstraction can be directly linked to changes in the early processing stage of visual information. Consonant with this interpretation, recent work in mice has elegantly reported how value governs a functional remapping in the sensory cortex by direct lateral OFC projections carrying RPE information (*Banerjee et al., 2020*), as well as by modulating the gain of neurons to irrelevant stimuli (*Liu et al., 2020*). While these considerations clearly point to a central role of the vmPFC and valuation in abstraction by controlling sensory representations, it remains to be investigated whether this effect results in more efficient *construction* of abstract representations, or in better *selection* of internal abstract models.

Given the complex nature of our design, there are some limitations to this work. For example, there isn't an applicable feature decoder to test actual feature representations (e.g. colour vs orientation), or the likelihood of one feature against another. In our task design, in every trial, all features

were used to define pacman characters. Furthermore, we did not have a localiser session in which features were presented in isolation (see Supplementary note two for further discussion). Future work could investigate how separate feature representations emerge on the path to abstractions, for example in the parietal cortex or vmPFC, and their relation to feature levels (e.g. for colour: red vs green) as reported here. We speculate that parallel circuits linking the prefrontal cortex and basal ganglia could track these levels of abstraction, possibly in hierarchical fashion (*Badre and Frank, 2012*; *Cortese et al., 2019*; *Haruno and Kawato, 2006*).

Some may point out that what we call 'Abstract RL' is, in fact, just attention-mediated enhanced processing. Yet, if top-down attention were the sole driver in Abstract RL, we contend that the pattern of results would have been different. For example, we would expect to see a marked increase in feature decodability in VC (*Guggenmos et al., 2015*; *Kamitani and Tong, 2005*). This was not the case here, with only a minimal, non-significant, increase. More importantly, the results of the decoded neurofeedback manipulation question this interpretation. Because decoded neurofeedback operates unconsciously (*Muñoz-Moldes and Cleeremans, 2020*; *Shibata et al., 2019*), value was added directly at the sensory representation level (limited to the targeted region), precluding alternative top-down effects. That is not to say that attention does not significantly mediate this type of abstract learning; however, we argue that attention is most likely an effector of the abstraction and valuation processes (*Krauzlis et al., 2014*). A simpler top-down attentional effect was indeed evident in the supplementary analysis comparing feature decoding in 'relevant' and 'irrelevant' cases (*Figure 5D*). Occipital regions displayed large effect sizes, irrespective of the learning strategy used to solve the task.

While valuation and abstraction appear tightly associated in reducing the dimensionality of the task space, what is the underlying mechanism? The degree of neural compression in the vmPFC has been shown to relate to features most predictive of positive outcomes, under a given goal (*Mack et al., 2020*). Similarly, the geometry of neural activity in generalisation may be key here. Neural activity in the PFC (and HPC) explicitly generates representations that are simultaneously abstract and high dimensional (*Bernardi et al., 2020*). An attractive view is that valuation may be interpreted as an abstraction in itself. Value could provide a simple and efficient way for the brain to operate on a dimensionless axis. Each point on this axis could index a certain task state, or even behavioural strategy, as a function of its assigned abstract value. Neuronal encoding of feature-specific value, or choice options, may help the system construct useful representations that can, in turn, inform flexible behavioural strategies (*Niv, 2019*; *Schuck et al., 2016*; *Wilson et al., 2014*).

In summary, this work provides evidence for a function of valuation that exceeds the classic view in decision-making and neuroeconomics. We show that valuation subserves a critical function in constructing abstractions. One may further speculate that valuation, by generating a common currency across perceptually different stimuli, may be an abstraction in itself, and that it is tightly linked to the concept of task states in decision-making. We believe this work not only offers a new perspective on the role of valuation in generating abstract thoughts, but also reconciles apparently disconnected findings in decision-making and memory literature on the role of the vmPFC. In this context, value is not a simple proxy of a numerical reward signal, but is better understood as a conceptual representation or schema built on-the-fly to respond to a specific behavioural demand. Thus, we believe our findings provide a fresh view of the invariable presence of value signals in the brain that play an important algorithmic role in development of sophisticated learning strategies.

## Materials and methods

### Participants

Forty-six participants with normal or corrected-to-normal vision were recruited for the main experiment (learning task). The sample size was chosen according to prior work and recommendations on model-based fMRI studies (*Lebreton et al., 2019*). Based on pilot data and the available scanning time in one session (60 min), we set the following conditions of exclusion: failure to learn the association in three blocks or more (i.e. reaching a block limit of 80 trials without having learned the association), or failure to complete at least 11 blocks in the allocated time. Eleven participants were removed based on the above predetermined conditions, 2 of which did not go past the initial practice stages. Additionally, two more participants were removed due to head motion artifacts. Thus,

33 participants (22.4 ± 0.3 y.o.; eight females) were included in the main analyses. Of these, 22 participants (22.2 ± 0.3 y.o.; four females) returned for the follow-up experiment, based on their individual availability. All results presented up to *Figure 5* are from the 33 participants who completed the learning task. All results pertaining to the neurofeedback manipulation are from the subset of 22 participants that were called back. *Figure 1—figure supplement 2* reports a behavioural analysis of the excluded participants to investigate differences in performance or learning strategy compared to the 33 included participants.

All experiments and data analyses were conducted at the Advanced Telecommunications Research Institute International (ATR). The study was approved by the Institutional Review Board of ATR with ethics protocol numbers 18–122, 19–122, 20–122. All participants gave written informed consent.

## Learning task

The task consisted of learning the fruit preference of pacman-like characters. These characters had three features, each with two levels (colour: green vs red, stripe orientation: horizontal vs vertical, mouth direction: left vs right). On each trial, a character composed of a unique combination of the three features was presented. The experimental session was divided into blocks, for each of which a specific rule directed the association between features and preferred fruits. There were always two relevant features and one irrelevant feature, but these changed randomly at the beginning of each block. Blocks could thus be of three types: CO (colour-orientation), CD (colour-direction), and OD (orientation-direction). Furthermore, to avoid a simple logical deduction of the rule after one trial, we introduced the following pairings. The four possible combinations of two relevant features with two levels were paired with the two fruits in both a symmetric or asymmetric fashion - 2x2 or 3x1. The appearance of the two fruits was also randomly changed at the beginning of each new block (see *Figure 1B,e*.g., green-vertical: fruit 1, green-horizontal: fruit 2, red-vertical: fruit 1, red-horizontal: fruit two *or* green-vertical: fruit 2, green-horizontal: fruit 2, red-vertical: fruit 1, red-horizontal: fruit 2).

Each trial started with a black screen for 1 s, following which the character was presented for 2 s. Then, while the character continued to be present at the centre of the screen, the two fruit options appeared below, to the right and left sides. Participants had 2 s to indicate the preferred fruit by pressing a button (right for the fruit to the right, left for vice versa). Upon registering a participant's choice, a coloured square appeared around the chosen fruit: green if the choice was correct, red otherwise. The square remained for 1 s, following which the trial ended with a variable ITI - bringing the trial to a fixed 8 s duration.

Participants were simply instructed that they had to learn the correct rule for each block, and the rule itself (relevant features + association type) was hidden. Each block contained up to 80 trials, but a block could end earlier if participants learned the target rule. Learning was measured as a set of correct trials (between 8 and 12, determined randomly in each block). Participants were instructed that each correct choice added one point, while incorrect choices did not alter the balance. They were further told that points obtained would be weighted by the speed of learning on that block. That is, the faster the learning, the greater the net worth of the points. The end of a block was explicitly signalled by presenting the reward obtained on the screen. Monetary reward was computed at the end of each block according to the formula:

$$R = A * \left( \frac{\sum pts}{\sum tr} \right) - \left( \sum tr - mcs \right) * a \tag{1}$$

where $R$ is the reward obtained in that block, $A$ the maximum available reward (150JPY), $\sum pts$ the sum of correct responses, $\sum tr$ the number of trials, $mcs$ the maximum length of a correct strike (12 trials), and $a$ is a scaling factor ($a$ = 1.5). This formula ensures time-dependent decay of the reward that approximately follows a quadratic fit. In case participants completed a block in less than 12 trials, if the amount was larger than 150JPY, it was rounded to 150JPY.

The maximum terminal monetary reward over the whole session was set at 3,000 JPY. On average, participants earned 1251 ± 46 JPY (blocks in which participants failed to learn the association within the 80-trial limit were not rewarded). For each experimental session, there was a sequence of 20 blocks that was pre-generated pseudo-randomly, and on average, participants completed 13.6 ±

0.3 blocks. In the post-neurofeedback behavioural test, all participants completed 16 blocks, 8 of which had the targeted feature as relevant, while in the other half the targeted feature was irrelevant. The order was arranged pseudo-randomly such that in both halves of the session there were four blocks of each type. In the post-neurofeedback behavioural session, all blocks had only asymmetric pairings with preferred fruits.

For sessions done in the MR scanner, participants were instructed to use their dominant hand to press buttons on a dual-button response pad to register their choices. Concordance between responses and buttons was indicated on the display, and importantly, randomly changed across trials to avoid motor preparation confounds (i.e. associating a given preference choice with a specific button press).

The task was coded with PsychoPy v1.82.01 (*Peirce, 2008*).

## Computational modelling part 1: mixture-of-experts RL model

We built on a standard RL model (*Sutton and Barto, 1998*) and prior work in machine learning and robotics to derive the mixture-of-experts architecture (*Doya et al., 2002*; *Jacobs et al., 1991*; *Sugimoto et al., 2012*). In this work, the mixture-of-experts architecture is composed of several 'expert' RL modules, each tracking a different representational space, and each with its own value function. In each trial, the action selected by the mixture-of-experts RL model is given by the weighted sum of the actions computed by the experts. The weight reflects the responsibility of each expert, which is computed from the SoftMax of the squared prediction error. In this section we define the general mixture-of-expert RL model, and in the next section we define each specific expert, based on the task-state representations being used.

Formally, the mixture-of-expert RL model global action is defined as:

$$A_t = \sum_{j=1}^{N} \lambda_t^j a_t^j \tag{2}$$

where $N$ is the number of experts, $\lambda$ the responsibility signal, and $a$ the action selected by the $j$th-model. Thus, $\lambda$ is defined as:

$$\lambda_t^j = exp\left(-\frac{\bar{RPE}_{t-1}^j}{v}\right) \Big/ \left(\sum_{k=1}^{N}\left\{exp\left(-\frac{\bar{RPE}_{t-1}^k}{v}\right)\right\}\right) \tag{3}$$

where $N$ is the same as above, $v$ is the RPE variance. Expert uncertainty $\bar{RPE}_t$ is defined as:

$$\bar{RPE}_t^j = \gamma \bar{RPE}_{t-1}^j + (1-\gamma)\left(RPE_t^j\right)^2 \tag{4}$$

where $\gamma$ is the forgetting factor that controls the strength of the impact of prior experience on the current uncertainty estimate. The most recent RPE is computed as:

$$RPE_t^j = O - Q^j(S_t, A_t) \tag{5}$$

where $O$ is the outcome (reward: 1, no reward: 0), $Q$ is the value function, $S$ the state for the current expert, and $A$ is the global action computed with *Equation (2)*. The update to the value function can therefore be computed as:

$$\Delta Q^j(S_t, A_t) = \lambda_t^j \alpha RPE_t^j \tag{6}$$

where $\lambda$ is the responsibility signal computed with *Equation (3)*, $\alpha$ is the learning rate (assumed equal for all experts), and *RPE* is computed with *Equation (5)*. Finally, for each expert, the action $a$ at trial $t$ is taken as the argmax of the value function, as follows:

$$a_t^j = argmax\left[Q^j(S_t, a)\right] \tag{7}$$

where $Q$ is the value function, $S$ the state at current trial, and $a$ the two possible actions.

Hyperparameters estimated through likelihood maximisation were the learning rate $\alpha$, the forgetting factor $\gamma$, and the RPE variance $v$.

## Computational modelling part 2: Feature RL and Abstract RLs

Each (expert) RL algorithm is built on a standard RL model (*Sutton and Barto, 1998*) to derive variants that differ in the number and type of states visited. Here, a state is defined as a combination of features. Feature RL has $2^3$ = eight states, where each state was given by the combination of all three features (e.g. colour, stripe orientation, mouth direction: green, vertical, left). Abstract RL is designed with $2^2$ = four states, where each state was given by the combination of two features.

Note that the number of states does not change for different blocks, only features used to determine them. These learning models create individual estimates of how participant action-selection depended on features they attended and their past reward history. Both RL models share the same underlying structure and are formally described as:

$$Q(s,a) \leftarrow Q(s,a) + \alpha(r - Q(s,a)) \tag{8}$$

where $Q(s,a)$ in *Equation (8)* is the value function of selecting either fruit-option $a$ for packman-state $s$. The value of the action selected in the current trial is updated based on the difference between the expected value and the actual outcome (reward or no reward). This difference is called the reward prediction error (RPE). The degree to which this update affects the expected value depends on the learning rate $\alpha$. For larger $\alpha$, more recent outcomes will have a strong impact. On the contrary, for small $\alpha$ recent outcomes will have little effect on expectations. Only the value function of the selected action, which is state-contingent in *Equation (8)*, is updated. Expected values of the two actions are combined to compute the probability $p$ of predicting each outcome using a Soft-Max (logistic) choice rule:

$$P_{s_i,A} = 1/(1 + exp(-\beta(Q(s_i,a_1) - Q(s_i,a_2)))) \tag{9}$$

The greediness hyperparameter $\beta$ controls how much the difference between the two expected values for $a_1$ and $a_2$ actually influence choices.

Hyperparameters estimated through likelihood maximisation were the learning rate $\alpha$, and the greediness (inverse temperature) $\beta$.

## Procedures for model fitting, simulations, and hyperparameter recovery

Hierarchical Bayesian Inference (HBI) was used to fit the models to participant behavioural data, enabling precise estimates of hyperparameters at the block level for each participant (*Piray et al., 2019*). Hyperparameters were selected by maximising the likelihood of estimated actions, given the true actions. For the mixture-of-experts architecture, we fit the model on all participants block-by-block to estimate hyperparameters at the single-block and single-participant level. For the subsequent direct comparison between Feature RL and Abstract RL models, we used HBI for concurrent model fitting and comparison at the single-block and single-participant basis. The model comparison provided the likelihood that each RL algorithm best explained participants' choice data. That is, it was a proxy to whether learning followed a Feature RL or Abstract RL strategy. Because the fitting was done block-by-block, with a hierarchical approach considering all participants, blocks were first sorted according to their lengths, from longer to shorter, at the participant level. This ensured that each block of a given participant was at the most similar to the blocks of all other participants, thus avoiding unwanted effects in the fitting due to block length. The HBI procedure was then implemented on all participant data, proceeding block-by-block.

We also simulated model action-selection behaviours to benchmark their similarity to human behaviour, and in the case of Feature RL vs Abstract RL comparisons, to additionally compare their formal learning efficiency. In the case of the mixture-of-experts RL architecture, we simply used estimated hyperparameters to simulate 45 artificial agents, each completing 100 blocks. The simulation allowed us to compute, for each expert RL module, the mean responsibility signal, and the mean expected value across all states for the chosen action. Additionally, we also computed the learning speed (time to learn the rule of a block) and compared it with the learning speed of human participants.

In the case of the simple Feature RL and Abstract RL models, we added noise to the state information in order to have a more realistic behaviour (from the perspective of human participants). In

the empirical data, the action (fruit selection) in the first trial of a new block was always chosen at random because participants did not have access to the appropriate representations (states). In later trials, participants may have followed specific strategies. For model simulations, we simply assumed that states were corrupted by a decaying noise function:

$$n_t = n_0 \left(1/t^{1/rte}\right) \qquad (10)$$

where $n_t$ is the noise level at trial $t$, $n_0$ the initial noise level (randomly drawn from a uniform distribution within the interval [0.3 0.7]), and $rte$ was the decay rate, which was set to 3. This meant that in early trials in a block, there was a higher chance of basing the action on the wrong representation (at random), rather than following the appropriate value function. Actions in later trials had a decreasing probability of being chosen at random. This approach is a combination of the classic ε-greedy policy and the standard SoftMax action-selection policy in RL. Hyperparameter values were sampled from obtained participant data maximum likelihood fits. We simulated 45 artificial agents solving 20 blocks each. The procedure was repeated 100 times for each block with new random seeds. We used two metrics to compare the efficiency of the two models: learning speed (same as above, the time to learn the rule of a block), as well as the fraction of failed blocks (blocks in which the rule was not learned with the 80-trials limit).

We performed a parameter recovery analysis for the simple Feature RL and Abstract RL models based on data from the main experiment. Parameter recovery analysis was done in order to confirm that fitted hyperparameters had sensible values and that the models themselves were a sensitive description of human choice behaviour (*Palminteri et al., 2017*). Using the same noisy procedure described above, we generated one more simulated dataset, using the exact blocks that were presented to the 33 participants. The blocks from simulated data were then sorted according to their length, and the hyperparameters α and β were fitted by maximising the likelihood of the estimated actions, given the true model actions. We report in *Figure 3—figure supplement 3* the scatter plot and correlation between hyperparameters estimated from participant data and recovered hyperparameters values, showing good agreement, notwithstanding the noise in the estimation.

For data from the behavioural session after multivoxel neurofeedback, blocks were first categorised as to whether the targeted feature was relevant or irrelevant to the rule of a given block. We then applied the HBI procedure as described above to all participants, with all blocks of the same type (e.g. targeted feature relevant) ordered by length. This allowed us to compute, based on whether the targeted feature was relevant or irrelevant, the difference in frequency between the models. We resampled with replacement to produce distributions of mean population bias for each block type.

## fMRI scans: acquisition and protocol

All scanning sessions employed a 3T MRI scanner (Siemens, Prisma) with a 64-channel head coil in the ATR Brain Activation Imaging Centre. Gradient T2*-weighted EPI (echoplanar) functional images with blood-oxygen-level-dependent (BOLD)-sensitive contrast and multi-band acceleration factor six were acquired (*Feinberg et al., 2010*; *Xu et al., 2013*). Imaging parameters: 72 contiguous slices (TR = 1 s, TE = 30 ms, flip angle = 60 deg, voxel size = $2 \times 2 \times 2$ mm$^3$, 0 mm slice gap) oriented parallel to the AC-PC plane were acquired, covering the entire brain. T1-weighted images (MP-RAGE; 256 slices, TR = 2 s, TE = 26 ms, flip angle = 80 deg, voxel size = $1 \times 1 \times 1$ mm$^3$, 0 mm slice gap) were also acquired at the end of the first session. For participants who joined the neurofeedback training sessions, the scanner was realigned to their head orientations with the same parameters for all sessions.

## fMRI scans: standard and parametric general linear models

BOLD-signal image analysis was performed with SPM12 [http://www.fil.ion.ucl.ac.uk/spm/], running on MATLAB v9.1.0.96 (r2016b) and v9.5.0.94 (r2018b). fMRI data for the initial 10 s of each block were discarded due to possible unsaturated T1 effects. Raw functional images underwent realignment to the first image of each session. Structural images were re-registered to mean EPI images and segmented into grey and white matter. Segmentation parameters were then used to normalise (MNI) and bias-correct the functional images. Normalised images were smoothed using a Gaussian kernel of 7 mm full-width at half-maximum.

GLM1: regressors of interest included 'High value', 'Low value' (trials were labelled as such based on the median split of the trial-by-trial expected value for the chosen option computed from the best fitting algorithm - Feature RL or Abstract RL), 'Feature RL', 'Abstract RL' (trials were labelled as such based on the best fitting algorithm at the block level). For all, we generated boxcar regressors at the beginning of the visual stimulus (character) presentation, with duration 1 s. Contrast of [1 -1] or [−1 1] were applied to the regressors 'High value' - 'Low value', and 'Feature RL' - 'Abstract RL'. Specific regressors of no interest included the time in the experiment: 'early' (all trials falling within the first half of the experiment), and 'late' (all trials falling in the second half of the experiment). The early/late split was done according to the total number of trials: taking as 'early', trials from the first block onward, adding blocks until the trial sum exceeded the total trials number divided by two.

GLM2 (PPI): the seed was defined as a sphere (radius = 6 mm) centred around the individual peak voxel from the 'High value' > 'Low value' group-level contrast, within the vmPFC (peak coordinates [2 50 -10], radius 25 mm). The ROI mask was defined individually to account for possible differences among participants. Two participants were excluded from this analysis, because they did not show a significant cluster of voxels in the bounding sphere (even at very lenient thresholds). The GLM for the PPI included three regressors (the PPI, the mean BOLD signal of the seed region, and the psychological interaction), as well as nuisance regressors described below.

For all GLM analyses, additional regressors of no interest included a parametric regressor for reaction time, regressors for each trial event (fixation, fruit options presentation, choice, button press [left, right], ITI), block regressors, the six head motion realignment parameters, framewise displacement (FD) computed as the sum of the absolute values of the derivatives of the six realignment parameters, the TR-by-TR mean signal in white matter, and the TR-by-TR mean signal in cerebrospinal fluid.

Second-level group contrasts from all models were calculated as one-sample t-tests against zero for each first-level linear contrast. Statistical maps were z-transformed, and then reported at a threshold level of *P(fpr)* < 0.001 (*Z* > 3.09, false positive control meaning cluster forming threshold), unless otherwise specified. Statistical maps were projected onto a canonical MNI template with MRIcroGL [https://www.nitrc.org/projects/mricrogl/] or a glassbrain MNI template with Nilearn 0.7.1 [https://nilearn.github.io/index.html].

## fMRI scans: pre-processing for decoding

As above, the fMRI data for the initial 10 s of each run were discarded due to possible unsaturated T1 effects. BOLD signals in native space were pre-processed in MATLAB v7.13 (R2011b) (MathWorks) with the mrVista software package for MATLAB [http://vistalab.stanford.edu/software/]. All functional images underwent 3D motion correction. No spatial or temporal smoothing was applied. Rigid-body transformations were performed to align functional images to the structural image for each participant. One region of interest (ROI), the HPC, was anatomically defined through cortical reconstruction and volumetric segmentation using the Freesurfer software [http://surfer.nmr.mgh.harvard.edu/]. Furthermore, VC subregions V1, V2, and V3 were also automatically defined based on a probabilistic map atlas (*Wang et al., 2015*). The vmPFC ROI was defined as the significant voxels from the GLM1 'High value' > 'Low value' contrast at *p*(fpr) < 0.0001, within the OFC. All subsequent analyses were performed using MATLAB v9.5.0.94 (r2018b). Once ROIs were individually identified, time-courses of BOLD signal intensities were extracted from each voxel in each ROI and shifted by 6 s to account for the hemodynamic delay (assumed fixed). A linear trend was removed from time-courses, which were further z-score-normalised for each voxel in each block to minimise baseline differences across blocks. Data samples for computing individual feature identity decoders were created by averaging BOLD signal intensities of each voxel over two volumes, corresponding to the 2 s from stimulus (character) onset to fruit options onset.

## Decoding: multivoxel pattern analysis (MVPA)

All ROI-based MVP analyses followed the same procedure. We used sparse logistic regression (SLR) (*Yamashita et al., 2008*), to automatically select the most relevant voxels for the classification problem. This allowed us to construct several binary classifiers (e.g. feature id.: colour - red vs green, stripes orientation - horizontal vs vertical, mouth direction - right vs left).

Cross-validation was used for each MVP analysis to evaluate the predictive power of the trained (fitted) model. In the primary analysis (reported in *Figure 5C*), cross-validation was done with a leave-one-run-out scheme, whereby each run was iteratively held out as a test set, and all other runs were used for training of the algorithm. The final accuracy was taken as the averaged accuracy across the runs. This approach is generally used because there may be subtle differences across runs: holding out one run as a test ensures higher generalizability of the results while avoiding within-run information leaks. Yet, because of the nature of our task and experiment, the leave-one-run-out cross-validation leads to other confounds due to varying number of training trials across classes (e.g. colour red vs green) or conditions (Feature RL vs Abstract RL blocks). To control for this idiosyncratic feature of our design, we performed a second cross-validation scheme. Here, we first merged the data from all blocks for each condition, and then computed the lowest bound of trial number from the minority class across conditions (e.g. if Feature RL had 128 trials labelled as 'green', and 109 as 'red', while Abstract RL had 94 trials labelled as 'green', and 99 labelled as 'red'; then the minority class lowest bound was 94). In each fold (N folds = 20), a number of trials equivalent to 80% of the minority class lowest bound was assigned to the training set from each class, and the remaining trials to the test set. The training samples were randomly chosen in each fold. Furthermore, for all MVP analysis, SLR classification was optimised by using an iterative approach (*Hirose et al., 2015*) In each fold of the cross-validation, the feature-selection process was repeated 10 times. In each iteration, selected features (voxels) were removed from pattern vectors, and only features with unassigned weights were used for the next iteration. At the end of the cross-validation, test accuracies were averaged for each iteration across folds, in order to evaluate accuracy at each iteration. The number of iterations yielding the highest classification accuracy was then used for the final computation. Results (*Figure 5C*, *Figure 5—figure supplement 3*) report the cross-validated average of the best yielding iteration.

For the multivoxel neurofeedback experiment, we used the entire dataset to train the classifier in VC. Thus, each classifier resulted in a set of weights assigned to the selected voxels. These weights could be used to classify any new data sample and to compute a likelihood of its belonging to the target class.

## Real-time multivoxel neurofeedback and fMRI pre-processing

As in previous studies (*Cortese et al., 2017*; *Cortese et al., 2016*; *Shibata et al., 2011*), during the multivoxel neurofeedback manipulation, participants were instructed to modulate their brain activity, in order to enlarge a feedback disc and maximise their cumulative reward. Brain activity patterns measured through fMRI were used in real time to compute the feedback score. Unbeknownst to participants, the feedback score, ranging from 0 to 100 (empty to full disc), represented the likelihood of a target pattern occurring in their brains at measurement time. Each trial started with an induction period of 6 s, during which participants viewed a cue (a small grey circle) that instructed them to modulate their brain activity. This period was followed by a 6 s rest interval, and then by a 2 s feedback, during which the disc appeared on the screen. Finally, each trial ended with a 6 s inter-trial interval (ITI). Each block was composed of 12 trials, and one session could last up to 10 blocks (depending on time availability). Participants did two sessions on consecutive days. Within a session, the maximum monetary bonus was 3000 JPY.

Feedback was calculated through the following steps. In each block, the initial 10 s of fMRI data were discarded to avoid unsaturated T1 effects. First, newly measured, whole-brain functional images underwent 3D motion correction using Turbo BrainVoyager (Brain Innovation, Maastricht, Netherlands). Second, time-courses of BOLD signal intensities were extracted from each of the voxels identified in the decoder analysis for the target ROI (VC). Third, the time-course was detrended (removal of linear trends), and z-score-normalised for each voxel using BOLD signal intensities measured up to the last point. Fourth, the data sample to calculate the target likelihood was created by taking the average BOLD signal intensity of each voxel over the 6 s (6 TRs) 'induction' period as in previous studies (*Cortese et al., 2016*; *Shibata et al., 2011*). Finally, the likelihood of each feature level (e.g. right vs left mouth direction) being represented in the multivoxel activity pattern was calculated from the data sample using weights of the constructed classifier.

## Data and code availability

Behavioural data, group-level maps of brain activation, and custom code used to generate results and figures are available at https://github.com/BDMLab/Cortese_et_al_2021 copy archived at swh:1:rev:3ac5090fe0af132364bbf92b9b0dff95919d60ee (*Cortese et al., 2021*).

## Acknowledgements

We thank Kaori Nakamura, Yasuo Shimada for experimental assistance; Drs. Hakwan Lau, Jessica Taylor for helpful comments on previous versions of this manuscript. **Funding**: JST ERATO (Japan, grant number JPMJER1801) to AC, AY, and MK; AMED (Japan, grant number JP18dm0307008) to AC and MK; the Chilean National Agency for Research and Development (ANID)/Scholarship Program/DOCTORADO BECAS CHILE/2017–72180193 to PS; the Royal Society and Wellcome Trust, Henry Dale Fellowship (102612/A/13/Z) to BDM.

## Additional information

### Funding

| Funder | Grant reference number | Author |
| --- | --- | --- |
| Japan Science and Technology Agency | JPMJER1801 | Aurelio Cortese Mitsuo Kawato |
| Japan Agency for Medical Research and Development | JP18dm0307008 | Aurelio Cortese Mitsuo Kawato |
| Chilean National Agency for Research and Development | 72180193 | Pradyumna Sepulveda |
| Wellcome Trust | 102612/A/13/Z | Benedetto De Martino |

The funders had no role in study design, data collection and interpretation, or the decision to submit the work for publication.

### Author contributions

Aurelio Cortese, Conceptualization, Resources, Data curation, Software, Formal analysis, Supervision, Funding acquisition, Validation, Investigation, Visualization, Methodology, Writing - original draft, Project administration, Writing - review and editing; Asuka Yamamoto, Software, Formal analysis, Investigation, Methodology; Maryam Hashemzadeh, Software, Formal analysis, Writing - review and editing; Pradyumna Sepulveda, Conceptualization, Software, Investigation, Writing - review and editing; Mitsuo Kawato, Conceptualization, Formal analysis, Supervision, Funding acquisition, Writing - original draft, Writing - review and editing; Benedetto De Martino, Conceptualization, Formal analysis, Supervision, Validation, Writing - original draft, Project administration, Writing - review and editing

### Author ORCIDs

Aurelio Cortese ![ORCID] https://orcid.org/0000-0003-4567-0924
Pradyumna Sepulveda ![ORCID] https://orcid.org/0000-0003-0159-6777
Benedetto De Martino ![ORCID] https://orcid.org/0000-0002-3555-2732

### Ethics

Human subjects: All experiments and data analyses were conducted at the Advanced Telecommunications Research Institute International (ATR). The study was approved by the Institutional Review Board of ATR with ethics protocol numbers 18-122, 19-122, 20-122. All participants gave written informed consent.

### Decision letter and Author response

Decision letter https://doi.org/10.7554/eLife.68943.sa1
Author response https://doi.org/10.7554/eLife.68943.sa2

## Additional files

**Supplementary files**

• Transparent reporting form

### Data availability

All data generated or analysed during this study are included in the manuscript and supporting files. Source data files have been provided for Figures 1-6. Behavioural data, group-level maps of brain activation, and custom code used to generate results and figures are available at https://github.com/BDMLab/Cortese_et_al_2021, copy archived at https://archive.softwareheritage.org/swh:1:rev:3ac5090fe0af132364bbf92b9b0dff95919d60ee.

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

# Appendix 1

## Supplementary note 1
### Target and control regressors for main GLM

On average Abstract RL blocks tended to be later blocks (*Figure 3F–G*, *Figure 3—figure supplement 2A*), and to be associated with a small but significantly higher ratio of correct to incorrect responses (*Figure 5—figure supplement 1*). Moreover, although Abstract RL blocks were associated with higher expected value compared with Feature RL blocks (*Figure 3—figure supplement 2C*), at the trial level value (high / low) and learning strategy (Feature RL or Abstract RL) were uncorrelated (*Figure 5—figure supplement 2A*), thus confirming the regressors' orthogonality. The main analysis that was used for the value contrast and for the strategy contrast thus included regressors for 'early', 'late', 'High value', 'Low value', 'Feature RL', 'Abstract RL', such that the GLM explicitly controlled for the idiosyncratic features of the task. Other regressors of no interest were motion parameters, mean white matter signal, mean cerebro-spinal fluid signal, block, constant.

## Supplementary note 2
### Levels of multivoxel fMRI neurofeedback

It is worth noting that the neurofeedback procedure targeted one feature's level, for example red colour, rather than colour overall. One might wonder why this approach would work nevertheless? Given previous work with fMRI-based decoded neurofeedback (1), the main driver of the effect was most likely due to change in processing in VC, leading to increased functional representation of task features also in PFC (particularly, in vmPFC). Because in the current work feature levels were intrinsically coupled in task space, for example if red-horizontal corresponded to fruit 1, then green-vertical too, enhanced processing of red should also directly influence the paired colour.

