## [Decision Letter]

**Acceptance summary:**

This study combines a novel behavioral task, reinforcement learning modeling, functional imaging, and neurofeedback to show that learning to focus on what information is important for predicting choice outcomes (i.e., "abstraction") is guided by value signals. Because "abstraction" is a key process underlying flexible behavior, understanding its neural and computational basis is of major importance for cognitive neuroscience.

**Decision letter after peer review:**

Thank you for submitting your article "Value signals guide abstraction during learning" for consideration by *eLife*. Your article has been reviewed by 3 peer reviewers, and the evaluation has been overseen by a Reviewing Editor and Michael Frank as the Senior Editor. The following individual involved in review of your submission has agreed to reveal their identity: Alireza Soltani (Reviewer #1).

Essential revisions:

All three reviewers agreed that your study is well-conducted and the results convincing. However, they also had specific questions and suggestions for improvement. Below is a list of 'essential' comments that we would expect you to address in a revised version of your manuscript. The reviewers also made additional comments in the individual critiques, which we would encourage you to consider when preparing your revision.

1. There were several questions about the modeling. Please add a formal model comparison (accounting for model complexity) and show how much the mixture of expert RL model improves the fit over purely abstract and/or feature RL models. In addition, please show the responsibility values for the mixture model. This information, in addition to 'mean expected value', is needed to draw conclusions on the importance of feature and abstract RL models.

2. Please add an analysis of the behavior of excluded subjects. Do they adopt a different strategy and that is why they could not learn fast/accurately enough?

3. Does VC-vmPFC coupling predict the abstraction level? This connection seems to be as important to the authors' claims as the discussed relationship between VC-vmPFC coupling and learning speed (Figure 4C).

4. Please discuss issues around efficiency and plausibility that result from running 4 models simultaneously. That is, would it not be better if the brain only implemented the most complex algorithm instead of this algorithm in addition to the 3 simpler models?

5. Could it be that participants are simply better (more correct choices) in the abstract blocks (which presumably are also the later blocks)? If so, what does that mean for the value contrast in vmPFC? DO they reflect performance or strategy?

6. Please plot performance separately for CO, CD and OD blocks as well as for 2x2 vs 3x1 blocks.

7. Please clarify how the variance over RPEs (v) was calculated.

8. Was the cross-validation done between runs? If not, if should be done between runs, if possible.

9. The specific difference between relevant and irrelevant features seems important. Please add Figure S6 into the main manuscript.

10. Please add the results of the neurofeedback experiment. Were participants successful at increasing the size of the disc? Was there a correlation between this success and subsequent performance on the association paradigm? Full results can be provided in the supplements but should be referenced in the main text.

*Reviewer #1:*

Overall, the question studied in this work is timely, interesting and important. More specifically, although previous modeling studies have been focused on explaining how humans and other animals can learn informative abstract representations at the behavioral level, the underlying neural mechanisms remained poorly understood. Cortese and colleagues performed modeling analyses using a mixture of experts RL that consist of abstract and feature RL models as well as behavioral analyses (analyses of choice in a learning task) to demonstrate that performance of human subjects in a multi-dimensional learning task depends on their adopted level of abstraction. Supporting their modeling and behavioral analyses, authors analyzed fMRI data to demonstrate that the connections between ventromedial prefrontal cortex (vmPFC), the brain area encoding value signals, and visual cortex (VC) can predict subjects' learning speed, which is an indicator for adoption of abstract representations. Lastly, to demonstrate the causal relationship between VC and adoption of abstract representations, authors used a multivoxel neurofeedback procedure and showed that artificially adding value to features in VC results in an increase in adoption of abstract representations.

Although provided analyses are thorough and results are convincing, further quantitative analyses could be included to strengthen the main claims of the study. More specifically, it is helpful to show that results from fitting the mixture of expert approach is fully consistent with analyses using purely Abstract and/or Feature RL models. Additionally, an analysis of excluded subjects' behavior is missing. This is important, because failure in performing the task could indicate alternative (but unsuccessful) representations adopted by some subjects.

Comments for the authors:

(1.1) Page 6-8 and Figure 2: I could have missed this, but authors don't seem to provide any formal comparison between the goodness-of-fit using the mixture of expert RL model and pure Feature RL or Abstract RL models. For example, a simple Abstract RL model with only the informative features could capture behavior of certain subjects. I asked this because, the mixture of expert RL model contains more parameters than the Feature and Abstract RL models. I wonder when accounting for the extra parameters, would the mixture of expert RL model still provide a better fit? Please clarify.

(1.2) Related to the previous point, if the mixture of expert RL model provides a better fit, how much of the captured variance is related to the Feature RL vs Abstract RL experts (e.g. Figure 3F, G)? Perhaps this could be answered by examining the weight assigned to each type of RL in this model (λ values).

(1.3) Authors don't show the values of responsibility signals in the mixture of expert model. This information, in addition to 'mean expected value', is needed to draw conclusions on the importance of Feature and Abstract RL models.

2) I feel that the decoding analysis can be further improved. For example, do authors see any changes happen as a result of experience in the task? Also, a relevant reference is a study by Oemisch et al., Nat. Comm 2019, in which the prevalence of feature encoding neurons is examined.

3) How did the excluded subjects perform the task? Do they adopt a different strategy and that is why they could not learn fast/accurately enough? For example, did they learn about the value of different features and combine these values to make decisions (as in feature-based RL in Farashahi et al., 2017 or Farashahi et al., 2020, which is different from Feature RL and closer to Abstract RL)? Please comment.

4) Does VC-vmPFC coupling predict the abstraction level? This connection seems to be as important to authors' claims as the discussed relationship between VC-vmPFC coupling and learning speed (Figure 4C).

*Reviewer #2:*

Cortese and colleagues report two experiments in which human subjects made choices based on cues that had three distinct visual features. Only two of the three visual features were needed to make a correct choice. Hence participants could safely ignore one feature and learn based only on the two relevant features (a process the authors call abstraction). The authors modelled how behaviour and ventromedial prefrontal cortex activity shifted from processing all features to only the two relevant features and sought to elucidate the role of feature valuation in this process. In a second experiment, the authors used a real-time neurofeedback approach to tag visual representations of features and showed how this feature valuation process shapes the feature selection described in Experiment 1.

Past research has investigated the process by which humans and other animals learn to attend relevant features during reinforcement learning (e.g., Niv et al., 2015; Leong et al., 2017). These studies have outlined how reward shapes which features we pay attention to, and how attention shapes how we process reward. While a true account of how the process of "abstraction" might occur is still outstanding in my opinion (see below), this study adds some important insights about this process. A main point is that the authors show changing representations of features directly in vmPFC, which co-occur and interact with values. They also provide insight into the unique roles vmPFC and the hippocampus might have in this process, and how vmPFC value signals interact with sensory areas.

One particularly interesting aspect is this study is the use of neurofeedback to achieve reward-tagging of visual representations. This approach is noteworthy as it does not require to pair reward with the visual features themselves, but rather with the occurrence of neural representations that reflect said features. The behavioural effects of this manipulation on later learning were impressively strong: if the task required to attend features that were tagged with reward, behaviour was guided more strongly by appropriate selective learning; if the task required to ignore the features that were previously tagged with reward, the learning process was unchanged. This suggests that the process of selecting relevant features during learning interacts with a neural mechanism that tracks the values associated with these features. This conclusion is also supported by the fact that the same brain area that tracked the expected values of the stimuli during the task, vmPFC, was modulated by participants' level of feature selection, i.e. abstraction.

One weakness of this study is that the mechanism of abstraction remains unclear. The authors use a mixture of experts architecture of 4 different RL models: one RL model that tries to learn the appropriate action as a function of all visual features of the cues, and 3 models that try to learn based on the possible subsets of only two features.

I have some concerns about this approach. One concern is that the modelling presumes that participants concurrently run all 4 RL models, and continuously decide which one is best. The whole purpose of using a lower dimensional model is that it is more efficient. Permanently using 4 models, including the highest dimensional one, seems to defy the purpose of why the search for a best model was initiated in the first place. Arguably, such a scheme would also not necessarily predict that vmPFC should come to selectively represent only the most relevant features, since the model that requires processing all features needs to be kept up to date. It also does not shed light on how participants could ever truly stop to pay attention to some features, as feature selection is only done by weighting the model with the lowest prediction errors relative to the variance most strongly in the action selection process. In other words: I am unsure if the manuscript presents a reasonable account how representations become transformed. Other models, which do not suffer from these shortcomings, such as a function approximation model, might have added important insights to this study. Ideally, the presented model could also explain another interesting observation made by the authors: that performance improves over blocks, even though the relevant features change. This probably reflects that participants might have learned something more global about the dimensionality of the relevant space, but such a learning process is not accounted for by the authors. On the positive side, while such a concurrent training of 4 models seems computationally inefficient, it is at least data efficient, as each experience is used to update all models at once. And, the mixture of experts approach may be considered a tool to investigate feature selection, rather than a cognitive model. This should be clarified in the manuscript.

Another weakness of the manuscript in my opinion is that the valuation process targeted in the neurofeedback experiment presupposes that visual features are predictive of reward. One important aspect of abstraction, however, is that they may not be, as the same feature could lead to different outcomes, for instance based on unobservable context.

Comments for the authors:

– It would be great to try to model how longer-term knowledge about rewarding features and dimensionality of the task influence performance. How does the change over blocks occur? How do biases, as introduced through the neurofeedback procedure, influence model selection in the mixture of experts approach?

– Figure 5: Could it be that participants are simply better (more correct choices) in the abstract blocks (which presumably are also the later blocks)? If so, wouldn't that mean that the contrast high-low value in vmPFC will necessarily be higher for abstract blocks, but it could reflect performance rather than strategy?

– It would be interesting to see performance separately for CO, CD and OD blocks as well as for 2x2 vs 3x1 blocks. Is there a difference between 2x2 vs 3x1 blocks?

– Please consider avoiding the word "predict" when reporting a regression analyses or other types of non-causal effects.

– I did not follow how the variance over RPEs (v) is calculated. It would be important to clarify that in the manuscript, and indicate how it changes as learning progresses.

Does a small variance imply that all models have similar RPEs? If so, I am not sure the statement that it is related to sharper model selection is the only way to view it. It seems it could also be related to more model similarity.

– Isn't the fact that the relevant AbRL has higher values and learns faster trivial, given the design of the task? Would there have been any possibility that these results would not have come out? If not, I believe all p values should be removed.

– It would be great to add the results from Figure S6 into the main manuscript. The specific different between relevant and irrelevant features seems important

– Decoding: was the cross validation done between runs? If not, if should be done between runs if possible

– Neurofeedback: can you provide more information about how good participants were, and how long the neurofeedback effect was presented in the later task blocks (did it diminish over time?).

*Reviewer #3:*

The authors of this study aimed to demonstrate that abstract representations occur during the course of learning and clarify the role of the vmPFC in this process. In a novel association learning paradigm it was shown that participants used abstract representations more as the experiment went on, and that these representations resulted in enhanced performance and confidence. Using decoded neurofeedback, (implicit) attention to certain features was reinforced monetarily and this led to these features being used more during the association task. They conclude that top down control (vmPFC control of sensory cortices) guides the use of abstract representations.

The strengths of this paper include an objective, model based assessment of reinforcement learning, a strong and simple experimental paradigm incorporating variable stopping criteria, and the incorporation of decoded neurofeedback to determine if these representations could be covertly reinforced and affect behavior.

The weaknesses include a small sample, the lack of subjective evaluation of strategies/learning, and the omission of neurofeedback learning results.

Overall the authors achieved their aims and the data supports their conclusions.

This work will be of significance to computational psychologists, those who study abstraction and decision making, and those interested in the role of the vmPFC. One exciting implication of this work is that the use of certain features can be reinforced via decoded neurofeedback.

Comments for the authors:

I am not an expert in computational methods, therefore my comments are largely restricted to the neurofeedback study. The neurofeedback task is well designed and the use of relevant and irrelevant features is a nice control condition. That the effects were only observed for relevant blocks and the finding of increased abstraction from the late blocks of the main experiment strengthens their conclusions regarding causality.

While this is not the main focus of the manuscript, a supplement should contain the results of the neurofeedback experiment. Were participants successful at increasing the size of the disc? Was there a correlation between this success and subsequent performance on the association paradigm?

6s of modulation seems short for neurofeedback studies, please justify this short modulation time.

Finally, I am curious as to whether subjects were interviewed regarding the strategies they were using during the association paradigm. Were they aware they were using abstraction?

[Editors' note: further revisions were suggested prior to acceptance, as described below.]

Thank you for resubmitting your work entitled "Value signals guide abstraction during learning" for further consideration by *eLife*. Your revised article has been evaluated by Michael Frank (Senior Editor) and a Reviewing Editor.

The manuscript has been improved but there are some remaining issues that need to be addressed, as outlined below:

1. Please revise your paper such that a casual reader will not erroneously take away that the MoE model presents a solid account of the data. Right now, the MoE is still mentioned in the abstract and also presented prominently in one of the main figures. You can leave the model in the manuscript if you want, but please further tone down any claims related to it.

2. It would be important to mention that some of the excluded subjects had good overall performance and the distribution of strategies was different among them.

*Reviewer #1:*

Authors have adequately and thoroughly addressed my concerns and questions. The only remaining concern I have is related to point # 1. Based on the presented results, it seems that MoE model does not provide the best fit of data. However, authors clearly mention and discuss this limitation in the revised manuscript. I have no further comments or concerns.

*Reviewer #2:*

I thank the authors for their thorough response to our previous concerns. I have two main concerns left:

1. The model comparison seems to refute the MoE model. At the same time, it seems clear that neither the FeRL nor AbRL model alone can truly capture participants behavior, since participants switch from one model to the other during the course of behavior. I think this should be made very clear in the paper, and I wonder how useful including the MoE model is.

My main reason is as follows: the core benefit of the MoE model, its ability to flexibly mix the two strategies, is seemingly not implemented in a way that reflects participants behavior. Would there be any way to improve the MoE models flexibility? The fact that it provides a "proof of concept that an algorithmic solution to arbitrate between representations / strategies exists" alone does not convince me, since the arbitration itself seems to not capture behavior and the pure existence of some algorithm is hardly surprising. In addition, there are the concerns about how realistic the MoE model is, which were raised under point 4.

I am also wondering whether the bad fit of the MoE model reflects how the fits were calculated: within each block, and then averaged (if I understood correctly)? Does that mean there was a new set of parameters per block? Have the authors tried to fit over the entirety of the experiment, using one set of parameters?

Relatedly, I believe that the change between strategies over time should be presented in one of the main figures, as this is an important point (e.g. by putting the rightmost graph from the Figure shown in the point by point response in the main paper).

2. I am also not fully convinced by the explanations about exclusions. The fact that the excluded subjects showed a different distribution of strategies should not serve as a reason for exclusion, since the purpose of the paper is to elucidate, in an unbiased manner, the distribution as it exists in the general population. The reported accuracy also does not seem very low for some participants. To me it seems that including the overall high performing subjects (with e.g. avg % correct > 70%) would provide a more unbiased sample.

---

## [Author Response]

Essential revisions:All three reviewers agreed that your study is well-conducted and the results convincing. However, they also had specific questions and suggestions for improvement. Below is a list of 'essential' comments that we would expect you to address in a revised version of your manuscript. The reviewers also made additional comments in the individual critiques, which we would encourage you to consider when preparing your revision.1. There were several questions about the modeling. Please add a formal model comparison (accounting for model complexity) and show how much the mixture of expert RL model improves the fit over purely abstract and/or feature RL models. In addition, please show the responsibility values for the mixture model. This information, in addition to 'mean expected value', is needed to draw conclusions on the importance of feature and abstract RL models.

As per your suggestion we have now included a formal model comparison. Using a hierarchical Bayesian approach (Piray et al. 2019), we performed concurrent fitting and comparison of all 3 models mixture-of-experts (MoE-RL), Feature RL (FeRL), Abstract RL (AbRL). We now show in supplementary (Figure 2—figure supplement 1) the model frequency (i.e., goodness of fit) in the sample. More specifically, for each block, we report the number of participants for which the MoE-RL, FeRL, or AbRL best explain their learning/choice data. Furthermore, Figure 2D in the main text now includes the responsibility values for the mixture model, in the same format as the ‘mean expected value’.

Note the model comparison shows that the MoE-RL in itself improves the fit over the purely abstract or feature RL only in very few instances. It is important to remember that FeRL and AbRL are much simpler models and that AbRL is an oracle model (i.e. the relevant dimension – unknown to the participant, is set by the experimenter). Our rationale in devising the MoE-RL model in the first part of the study was not to show that it is superior to the purely abstract or feature RL. MoE-RL was introduced as a proof of concept that an algorithmic solution in the arbitration between strategies is possible setting the stage for the second part of the work that focused on direct comparison of the two simpler models – Feature RL and Abstract RL (that have been used for all the remaining neuroimaging data analysis). We have now amended the text in the manuscript to make this clearer.

Main text (pp. 8, lines 212 – 227):

“The mixture-of-expert RL model revealed that participants who learned faster relied more on the best RL model value representations. […] Hence, we next sought to explicitly explain participant choices and learning according to either Feature RL or Abstract RL strategy.”

2. Please add an analysis of the behavior of excluded subjects. Do they adopt a different strategy and that is why they could not learn fast/accurately enough?

Of the 13 subjects that were excluded, for 2 we did not have recorded data as the experiment never went past the preliminary stage. As per your request we have now added the analysis of the 11 remaining excluded subjects (these also included 2 subjects that were removed for high head motion in the scanner). Briefly, excluded subjects displayed lower response accuracy in choosing the preferred fruit, as well as significantly higher (resp. lower) proportion of blocks labelled as Feature RL (resp. Abstract RL). Based upon these results, it would appear that excluded participants tended to remain ‘stuck’ in a non-optimal learning regime (Feature RL). These results are now reported in Figure 1—figure supplement 2 and referenced in the main text.

Main text (pp. 24, lines 673 – 675):

“Figure 1—figure supplement 2 reports a behavioural analysis of the excluded participants to investigate differences in performance or learning strategy compared to the 33 included participants.”

3. Does VC-vmPFC coupling predict the abstraction level? This connection seems to be as important to the authors' claims as the discussed relationship between VC-vmPFC coupling and learning speed (Figure 4C).

We agree with the reviewer that the VC-vmPFC coupling with learning speed suggests that a similar coupling might also exist with abstraction level. We have conducted the analysis suggested – we don’t find an effect that passes the statistical threshold but only a non-significant trend (robust linear regression, p = 0.065 one-sided). However, we should also highlight the exploratory nature of these between subjects’ correlations since our study was not optimised to detect between subjects' effects (which generally requires much larger n of subjects). We have now added the plot as Figure 4—figure supplement 2, and report the result in the main text and mentioned this caveat.

Main text (pp. 12, lines 331 – 335):

“The strength of the vmPFC – VC coupling showed a non-significant trend with the level of abstraction (N = 31, robust regression, slope = 0.013, t_29_ = 1.56, p = 0.065 one-sided, Figure 4—figure supplement 2). […] Therefore, future work is required to confirm or falsify this result.”

4. Please discuss issues around efficiency and plausibility that result from running 4 models simultaneously. That is, would it not be better if the brain only implemented the most complex algorithm instead of this algorithm in addition to the 3 simpler models?

As one of the reviewers astutely noticed, while the mixture-of-experts model is not the most computationally thrifty model, it is very data efficient (i.e., the same data points can be used to update multiple models / representations in parallel). This model was introduced in the manuscript not necessarily as the most realistic model but as a proof of concept of a cognitive architecture that can arbitrate between the abstract and feature-based learning strategies. This was to set the stage for comparing these 2 strategies in the neuroimaging data analysis that was the main scope of this work. We have now clarified this in the manuscript and added a formal model comparison in response to the first query. Note the model comparison shows that the MoE-RL in itself improves the fit over the purely abstract or feature RL only in very few instances. It is important to remember that FeRL and AbRL are much simpler models and that AbRL is an oracle model (i.e. the relevant dimension – unknown to the participant, is set by the experimenter). Our rationale in devising the MoE-RL model in the first part of the study was not to show that it is superior to the purely abstract or feature RL. MoE-RL was introduced as a proof of concept that a simple algorithmic solution in the arbitration between strategies is possible.

We agree with the reviewer that more work needs to be done to establish which is the actual computational basis in humans to select the correct strategy in each circumstance. We share their feeling that humans might (at least at the conscious level) engage with one hypothesis at a time. However, there is circumstantial evidence that multiple strategies might be computed in parallel but deployed one at a time (Domenech et al. 2014, Koechlin 2018). Given how little we know about how different algorithmic architectures to solve this kind of problem are implemented by the brain, we agree that these are important issues that need to be discussed. Making clear our goal to show that arbitration between feature and abstract learning can be achieved using a relatively simple algorithm (the MoE-RL) and then proceeded to characterise the neural underpinnings of these two types of learning (i.e. FeRL and AbRL). These points are also elaborated in the discussion.

Main text (pp. 18 – 19, lines 513 – 529):

“An interesting and open question concerns whether the brain uses abstract representations in isolation – operating in a hypothesis-testing regime – i.e., favouring the current best model; or whether representations may be used to update multiple internal models, with behaviour determined by their synthesis (as in the mixture-of-experts architecture). […] Future work will need to establish the actual computational strategy employed by the human brain, further examining how it may also vary across circumstances.

5. Could it be that participants are simply better (more correct choices) in the abstract blocks (which presumably are also the later blocks)? If so, what does that mean for the value contrast in vmPFC? DO they reflect performance or strategy?

The intuition is correct, in the Abstract blocks participants tended to make more correct choices on average (now reported in Figure 5—figure supplement 1), therefore Abstract RL blocks were associated with higher expected value compared with Feature RL blocks. As the reviewer correctly hinted this is probably due to the fact that abstract strategies were more frequent in late trials in which performance is usually higher. This small but significant result is now reported in Figure 3—figure supplement 2. Importantly for the GLM analysis at trial-by-trial level, value (high / low) and learning strategy (Feature RL or Abstract RL) were uncorrelated (Figure 5—figure supplement 2A) thus confirming the regressors’ orthogonality allowing us to include both regressors in the same GLM. We also included regressors for ‘early’, ‘late’.

To recapitulate we included regressors for ‘early’, ‘late’, ‘High value’, ‘Low value’, ‘Feature RL’, ‘Abstract RL’, such that the GLM explicitly controlled for the idiosyncratic features of the task.

We are therefore confident that our GLM was able to correctly disentangle the contribution of all these parameters on the neural signal.

We have updated the supplementary note 1 – we refer to it in the main text to better explain the idiosyncrasies of the task / conditions and their controls in the main GLM analysis. The text is reported here below.

Main text (pp. 12, lines 353 – 355):

“Having established that the vmPFC computes a goal-dependent value signal, we evaluated whether the activity level of this region was sensitive to the strategies that participants used. To do so, we used the same GLM introduced earlier, and estimated two new statistical maps from the regressors ‘Abstract RL’ and ‘Feature RL’ while controlling for idiosyncratic features of the task, i.e., high/low value and early/late trials (see Methods and Supplementary note 1).”

“Supplementary Note 1:

On average Abstract RL blocks tended to be later blocks (Figure 3F-G, Figure 3—figure supplement 2A), and to be associated with a slightly but significantly higher ratio of correct to incorrect responses (Figure 5—figure supplement 1). […] Other regressors of no interest were motion parameters, mean white matter signal, mean cerebro-spinal fluid signal, block, constant.”

6. Please plot performance separately for CO, CD and OD blocks as well as for 2x2 vs 3x1 blocks.

Thanks for the suggestion; we have done this. Figure S1 displays performance (learning speed) plotted separately for CO, CD, OD blocks, as well as for 2x2 and 3x1 blocks. There were no significant differences between these conditions.

7. Please clarify how the variance over RPEs (v) was calculated.

We apologize for our oversight – the variance over RPEs (v) was an hyperparameter estimated at the participant level, in each block. This has been clarified in the manuscript.

Main text (pp. 6, lines 177 – 179):

“Estimated hyperparameters (learning rate 𝛂, forgetting factor 𝛄, RPE variance 𝛎) were used to compute value functions of participant data, as well as to generate new, artificial choice data and value functions.”

8. Was the cross-validation done between runs? If not, if should be done between runs, if possible.

The cross-validation was done by repeatedly splitting the whole data in a training and test group, at random (N=20). We applied this procedure because the number of trials available for each class differed across conditions and best models (Feature RL and Abstract RL). For example, Feature RL may have had 128 trials labelled as ‘green’, 109 as ‘red’, while Abstract RL 94 as ‘green’, and 99 as ‘red’. We thus selected, in each fold, the number of trials representing 80% of the data in the condition with the lowest number of trials (in this example, 80% of 94). This procedure allowed us to avoid a situation in which different amounts of data are used to train the classifiers in different conditions, making comparisons or performance averages weaker to interpret. Nevertheless, since it has been rightfully pointed out that the procedure generally involves testing a classifier on data from a different run (leave-one-run-out cross-validation) such that even subtle differences across runs cannot be exploited by the algorithm.

Therefore, we have now implemented this procedure, recommended by the reviewer, as the primary analysis (reported in the new Figure 5C). Note that results from these two cross-validation approaches closely align, and we have now moved the original result to supplementary (Figure S10).

Main text (pp. 13, lines 380 – 389):

“We found that classification accuracy was significantly higher in Abstract RL trials compared with Feature RL trials in both the HPC and vmPFC (two-sided t-test, HPC: t_32_ = -2.37, p_(FDR)_ < 0.036, vmPFC: t_32_ = -2.51, p_(FDR)_ = 0.036, Figure 5C), while the difference was of opposite sign in VC (t_32_ = 1.61, p_(FDR)_ = 0.12, Figure 5C). […] A control analysis equating the number of training trials for each feature and condition replicated the original finding (Figure 5—figure supplement 3).”

Methods (pp. 32 – 33, lines 937 – 961):

“Cross-validation was used for each MVP analysis to evaluate the predictive power of the trained (fitted) model. […] Results (Figures 5C, Figure 5—figure supplement 3) report the cross-validated average of the best yielding iteration.”

9. The specific difference between relevant and irrelevant features seems important. Please add Figure S6 into the main manuscript.

We have inserted the previous Figure S6 into the main manuscript, as panels D and E in Figure 5.

10. Please add the results of the neurofeedback experiment. Were participants successful at increasing the size of the disc? Was there a correlation between this success and subsequent performance on the association paradigm? Full results can be provided in the supplements but should be referenced in the main text.

Participants were successful at increasing the size of the disc, with similar levels of performance attained in the first and second session. We also show the relationship between the success in inducing the target pattern and the subsequent behavioural effect. There was a significant tendency in positive correlation between the cumulative session-averaged amount of reward obtained during the NFB manipulation and the strength of the subsequent behavioural effect. Results are reported in the main text and in supplementary, Figure 6—figure supplement 1.

Main text (pp. 16, lines 463 – 466):

“Participants were successful at increasing the disk size in the neurofeedback task (Figure 6—figure supplement 1A-B). Furthermore, those who were more successful were also more likely to display larger increases in abstraction in the subsequent behavioural test (Figure 6—figure supplement 1C).”

[Editors' note: further revisions were suggested prior to acceptance, as described below.]

The manuscript has been improved but there are some remaining issues that need to be addressed, as outlined below:1. Please revise your paper such that a casual reader will not erroneously take away that the MoE model presents a solid account of the data. Right now, the MoE is still mentioned in the abstract and also presented prominently in one of the main figures. You can leave the model in the manuscript if you want, but please further tone down any claims related to it.

As per your suggestion we have toned down any claims related to the MoE model. To this end, we have amended the abstract, introduction, results and discussion. More specifically, we have removed any reference to the MoE model in the abstract and summary of results in the discussion to avoid giving the impression that the MoE model provides the best explanatory degree to the data. We further acknowledge in the discussion that with the current task the MoE model does not present a strong account of the data, and is not the main take-home message. Nevertheless, we have decided to keep the model in the manuscript and in Figure 2, as we still believe the MoE provides certain valuable pieces of information: (i) it provides a simple way to arbitrate between internal representations, (ii) it affords a data efficient approach (one data point can be used to update multiple strategies in parallel), (iii) it shows that participants who are more confident in their performance also have better selection of internal representations.

We report below excerpts of the main text that we have modified.

Abstract:

“Mixture-of-experts Reinforcement-learning algorithms revealed that, with learning, high-value abstract representations increasingly guided participant behaviour, resulting in better choices and higher subjective confidence.”

Main text, introduction (pp., lines):

“Reinforcement learning (RL) and mixture-of-experts (Jacobs et al., 1991; Sugimoto et al., 2012) modelling allowed us to track participant valuation processes and to dissociate their learning strategies (both at the behavioural and neural levels) based on the degree of abstraction.”

Main text, results (pp., lines):

Section title: “Mixture-of-experts reinforcement learning for Discovery of abstract representations”.

Main text, discussion (pp., lines):

“The ability to generate abstractions from simple sensory information has been suggested as crucial to support flexible and adaptive behaviours (Cortese et al., 2019; Ho et al., 2019; Wikenheiser and Schoenbaum, 2016). […] Of particular importance will be further examining how such strategies vary across circumstances (tasks, contexts, or goals).”

2. It would be important to mention that some of the excluded subjects had good overall performance and the distribution of strategies was different among them.

We agree that mentioning these aspects of the excluded subjects is important. We have thus added this information in the results, in the ‘Behavioural account of learning’ and ‘Behaviour shifts from Feature- to Abstraction-based reinforcement learning’ subsections.

To clarify, subjects were excluded independently of the distribution of strategies, which were computed at a later stage. The a priori criteria for exclusion were: failure to learn the association in 3 blocks or more (i.e., reaching a block limit of 80 trials without having learned the association), or failure to complete more than 10 blocks in the allocated time. These criteria were set to ensure a sufficient number of learning blocks for ensuing analyses.

To avoid any confusion, we have added this information to the main text, in results, subsection ‘Experimental design’.

Main text, results (pp., lines):

“Participants failing to learn the association in 3 blocks or more (i.e., reaching a block limit of 80 trials without having learned the association), and / or failing to complete more than 10 blocks in the allocated time, were excluded (see Methods). All main results reported in the paper are from the included sample of N = 33 participants.”

Main text, results (pp., lines):

“Excluded participants (see Methods) had overall lower performance (Figure 1 supplement 2), although some had comparable ratios correct.”

Main text, results (pp., lines):

“Given the lower learning speed in excluded participants, the distribution of strategies was also different among them, with a higher ratio of Feature RL blocks (Figure 3 supplement 4).”